# Scaling Up Bayesian Neural Networks with Neural Networks

## Abstract

Bayesian Neural Network (BNN) offers a principled and natural framework for proper uncertainty quantification in the context of deep learning. They address the typical challenges associated with conventional deep learning methods, such as data insatiability, ad-hoc nature, and susceptibility to overfitting. However, their implementation typically relies on Markov chain Monte Carlo (MCMC) methods that are characterized by their computational intensity and inefficiency in a high-dimensional space. To address this issue, we propose a novel calibration-Emulation-Sampling (CES) strategy to significantly enhance the computational efficiency of BNN. In this CES framework, during the initial calibration stage, we collect a small set of samples from the parameter space. These samples serve as training data for the emulator. Here, we employ a Deep Neural Network (DNN) emulator to approximate the forward mapping, i.e., the process that input data go through various layers to generate predictions. The trained emulator is then used for sampling from the posterior distribution at substantially higher speed compared to the original BNN. Using simulated and real data, we demonstrate that our proposed method improves computational efficiency of BNN, while maintaining similar performance in terms of prediction accuracy and uncertainty quantification.

## 1 Introduction

In recent years, Deep Neural Networks (DNN) have emerged as the predominant driving force in the field of machine learning and regarded as the fundamental tools for many intelligent systems (Cheng et al., 2018; LeCun et al., 1998; Sze et al., 2017). While DNN have demonstrated significant success in prediction tasks, they often struggle with accurately quantifying uncertainty. In recent years, there have been some attempts to address this issue. For example, the Ensemble Deep Learning method (Lakshminarayanan et al., 2017) aggregates predictions from multiple models to improve reliability and uncertainty estimates. While these methods represent important progress in the right direction, developing a principled and computationally efficient framework for Uncertainty Quantification (UQ) within the context of deep learning remains a significant challenge and active area of research. This is especially important in application areas, such as health sciences, where the level of uncertainty needs to be incorporated in the decision making process. Additionally, neural networks, due to their vulnerability to overfitting, can generate highly confident yet erroneous predictions (Su et al., 2019; Kwon et al., 2022). To address these issues, Bayesian Neural Network (BNN) (MacKay, 1992; Neal, 2012; Jospin et al., 2022) has emerged as an alternative to standard DNN, providing a transformative paradigm within the field of machine learning. By effectively incorporating Bayesian inference into the neural network framework, BNN provides a principled solution to these challenges. Their intrinsic ability to capture and quantify uncertainties in predictions establishes a robust foundation for decision-making under uncertainty. However, Bayesian inference in high-dimensional BNN poses significant computational challenges due to the inefficiency of traditional Markov Chain Monte Carlo (MCMC) methods. In fact, not only BNN, but almost all traditional Bayesian inference methods relying on MCMC techniques are known for their computational intensity and inefficiency when dealing with high-dimensional problems. Consequently, researchers have proposed various approaches to expedite the inference process (Welling & Teh, 2011b; Shahbaba et al., 2014; Ahn et al., 2014; Hoffman & Gelman, 2011; Beskos et al., 2017; Cui et al., 2016; Zhang et al., 2017a;b; 2018; Li et al., 2019a). Here, we focus on a state-of-the-art approach, called Calibration-Emulation-Sampling (CES) (Cleary et al., 2021), which has shown promising

results in large-dimensional UQ problems such as inverse problems (Lan et al., 2022a). CES involves the following three steps:

**(i)** Calibrate models to collect sample parameters and expensive forward evaluations for the emulation step

**(ii)** Emulate the parameter-to-data map using evaluations of forward models, and

**(iii)** Generate posterior samples using MCMC based on cheaper emulators.

This framework allows for reusing expensive forward evaluations and provides computationally efficient alternative to the standard MCMC procedure.

The standard CES method (Cleary et al., 2021) focuses on UQ in inverse problems and uses Gaussian Process (GP) models for the emulation component. GP models have a well-established history of application in emulating computer models (Currin et al., 1988), conducting uncertainty analyses (Oakley & O'Hagan, 2002), sensitivity assessments (Oakley & O'Hagan, 2004), and calibrating computer codes (Kennedy & O'Hagan, 2002; Higdon et al., 2004; O'Hagan, 2006). Despite their versatility, GP-based emulators are computationally intensive, with a complexity of $O(N^3)$, where $N$ is the sample size, using the squared-exponential kernel. Lower computational complexity can be achieved using alternative kernels (Lan et al., 2015) or various computational techniques (Liu et al., 2020; Bonilla et al., 2007; Gardner et al., 2018; Seeger et al., 2003). Nevertheless, scaling up GP emulators to high-dimensional problems remains a limiting factor. Furthermore, the prediction accuracy of GP emulators highly depends on the quality of the training data, emphasizing the importance of rigorous experimental design. To address these issues, Lan et al. (2022a) proposed an alternative CES scheme called Dimension Reduced Emulative Autoencoder Monte Carlo (DREAMC) method, which uses Convolutional Neural Networks (CNN) as emulator. DREAMC improved and scaled up the application of the CES framework for Bayesian UQ in inverse problems from hundreds of dimensions (with GP emulation) to thousands of dimensions (with NN emulation). Here, we adopt this approach and propose a new method, called Fast BNN (FBNN), for Bayesian inference in neural networks. We use DNN for the emulation component of our CES scheme. DNN has proven to be a powerful tool in a variety of applications and offers several advantages over GP emulation (Lan et al., 2022b; Dargan et al., 2020). It is computationally more efficient and suitable for high-dimensional problems. ~~Additionally, DNN exhibits greater robustness when dealing with variations in the training data.~~ The choice of DNN as an emulator enhances computational efficiency and flexibility.

Besides the computational challenges associated with building emulators, efficient sampling from posterior distributions in BNN also presents a significant challenge due to the high dimensionality of the target distribution. Traditional Metropolis-Hastings algorithms, typically defined on finite-dimensional spaces, encounter diminishing mixing efficiency as the dimensions increase (Gelman et al., 1997; Roberts & Rosenthal, 1998; Beskos et al., 2009). To overcome this inherent drawback, a novel class of 'dimension-independent' MCMC methods has emerged, operating within infinite-dimensional spaces (Beskos, 2014; Beskos et al., 2009; 2011; Cotter et al., 2013; Law, 2014; Beskos, 2014; Beskos et al., 2017). More specifically, we use the Preconditioned Crank-Nicolson (pCN) algorithm. The most significant feature of pCN is its dimension robustness, which makes it well-suited for high-dimensional sampling problems. The pCN algorithm is well-defined, with non-degenerate acceptance probability, even for target distributions on infinite-dimensional spaces. As a result, when pCN is implemented on a real-world computer in large but finite dimension $N$, the convergence properties of the algorithm are independent of $N$. This is in strong contrast to schemes such as Gaussian random walk Metropolis-Hastings and the Metropolis-adjusted Langevin algorithm, whose acceptance probability degenerates to zero as $N$ tends to infinity.

In summary, this paper addresses the critical challenges of UQ in high-dimensional BNN. By incorporating deep neural networks for emulation and leveraging the dimension-robust pCN algorithm for sampling, this research significantly enhances computational efficiency and scalability in Bayesian uncertainty quantification, offering a robust counterpart to DNN, and a scalable counterpart to BNN. Through extensive experiments, we demonstrate the feasibility and effectiveness of utilizing FBNN to accelerate Bayesian UQ in high-dimensional neural networks. The proposed method showcases remarkable computational efficiency, enabling scalable Bayesian inference in BNN with thousands of dimensions.

## 2    Related Methods

Various MCMC methods are employed to explore complex probability distributions for Bayesian inference. In this section, we discuss a set of MCMC methods used in our proposed FBNN model and discussed in the following sections.

Additionally, in this section we explore a variety of other methods utilized in our numerical experiments, which extend beyond MCMC frameworks. These include Ensemble Deep Learning for Neural Networks (Perrone & Cooper, 1995), BNNs with Variational Inference (Jaakkola & Jordan, 2000), BNNs leveraging Lasso Approximation (MacKay, 1992), Monte Carlo Dropout (MC-Dropout) (Gal & Ghahramani, 2016), Stochastic Weight Averaging-Gaussian (SWAG) (Maddox et al., 2019), and an Accelerated Hamiltonian Monte Carlo (HMC) model (Zhang et al., 2017c). These techniques provide a comprehensive spectrum for comparison with our FBNN model, serving as baseline methods for our analysis.

### 2.1    Hamiltonian Monte Carlo (HMC)

In general, MCMC methods represent a category of algorithms designed for sampling from a probability distribution (Andrieu et al., 2003). The fundamental principle involves building a Markov chain where the target distribution serves as its equilibrium distribution. Various algorithms exist for constructing such Markov chains, where the simplest is the Metropolis-Hastings algorithm (Metropolis et al., 1953). Metropolis-Hastings is a fundamental MCMC method used for obtaining a sequence of random samples from a probability distribution for which direct sampling is difficult (Chib & Greenberg, 1995; Robert et al., 1999).

Hamiltonian Monte Carlo is a special case of the Metropolis-Hastings algorithm, that incorporates Hamiltonian dynamics evolution and auxiliary momentum variables(Neal et al., 2011). Compared to using a Gaussian random walk proposal distribution in the Metropolis-Hastings algorithm, HMC reduces the correlation between successive sampled states by proposing moves to distant states which maintain a high probability of acceptance due to the approximate energy conserving properties of the simulated Hamiltonian dynamic. The reduced correlation means fewer Markov chain samples are needed to approximate integrals with respect to the target probability distribution for a given Monte Carlo error.

### 2.2    Stochastic Gradient Hamiltonian Monte Carlo (SGHMC)

As discussed earlier, HMC sampling methods provide a mechanism for defining distant proposals with high acceptance probabilities in a Metropolis-Hastings framework, enabling more efficient exploration of the state space than standard random-walk proposals. However, a limitation of HMC methods is the required gradient computation for simulation of the Hamiltonian dynamical system; such computation is infeasible in problems involving a large sample size. Stochastic Gradient Hamiltonian Monte Carlo (SGHMC) (Chen et al., 2014) addresses computational inefficiency by using a noisy but unbiased estimate of the gradient computed from a mini-batch of the data.

SGHMC is a valuable method for Bayesian inference, particularly when dealing with large datasets, as it leverages stochastic gradients and hyperparameter adaptation to efficiently explore high-dimensional target distributions.

### 2.3    Random Network Surrogate–HMC (RNS–HMC)

One common computational bottleneck for HMC for big data is repetitive evaluations of functions, their derivatives, geometric and statistical quantities. To alleviate this issue, in recent years several methods have been proposed to construct surrogate Hamiltonians. To this end, Random network surrogate–HMC (RNS–HMC) as an Accelerated HMC method introduced by Zhang et al. (2017c) uses random non-linear bases along with efficient learning algorithms to construct a surrogate functions that provides effective approximation of the probabilistic model based on the collective behavior of the large data. The goal is to explore and exploit the structure and regularity in parameter space for the underlying probabilistic model to construct an effective approximation of its geometric properties. To achieve this, RNS–HMC builds

---

**Algorithm 1** Preconditioned Crank-Nicolson Algorithm (pCN)

---
**Initialization:**
Choose an initial guess for the solution, $W^0$.
Set the iteration counter $n = 0$.
**while** not converged or maximum iterations not reached **do**
    **Implicit Step:**
    Solve the implicit part using Crank-Nicolson:

$$\frac{W^{n+1} - V^{n+1/2}}{\Delta t} = -\frac{1}{2}\left(A(W^{n+1}) + A(V^{n+1/2})\right)$$

    **Preconditioning:**
    Apply a preconditioning step.
    **Explicit Step:**
    Solve the explicit part:

$$\frac{V^{n+1} - W^{n+1}}{\Delta t} = -\frac{1}{2}\left(A(V^{n+1/2}) + A(W^{n+1})\right)$$

    **Acceptance Probability:**
    Compute the acceptance probability:

$$\alpha = \min\left(1, \exp(\Phi(W^n) - \Phi(V^{n+1}))\right)$$

    **Accept/Reject:**
    Generate a random number $u$ from a uniform distribution.
    **if** $u \leq \alpha$ **then**
        Set $W^{n+1} = V^{n+1}$.
    **else**
        Set $W^{n+1} = W^n$.
    **end if**
    **Update:**
    Increment the iteration counter: $n = n + 1$.
**end while**
**Output:**
The final solution is $W^N$ after $N$ iterations.

---

a surrogate function to approximate the target distribution using properly chosen random bases and an efficient optimization process.

The RNS–HMC process begins by identifying suitable bases that can capture the complex geometric properties of the data's parameter space. Through an optimization process, these bases are then used to form surrogate functions that stand in for the computationally expensive true Hamiltonian dynamics. Unlike traditional HMC, which requires repeated evaluation of the model and its derivatives, RNS-HMC leverages these surrogates to perform leapfrog integration steps, leads to reducing computational load. Later, Li et al. (2019b) extended this idea by using a neural network to directly approximate the gradient of the Hamiltonian, rather than using random bases for surrogate construction.

### 2.4 Preconditioned Crank-Nicolson (pCN)

Preconditioned Crank-Nicolson (Da Prato & Zabczyk, 2014) is a variant of MCMC incorporating a preconditioning matrix for adaptive scaling (Cotter et al., 2013). The key steps of the pCN algorithm are outlined in Algorithm 1.

In the given pCN Algorithm, the function $\Phi(W^n)$ represents a quantity associated with the current solution $W^n$ at the n-th iteration. The specific form and interpretation of $\Phi$ depend on the context of the problem being solved. It could be an energy function, a cost function, or any other relevant metric used to assess the quality or appropriateness of the current solution.

The pCN method involves re-scaling and random perturbation of the current state, incorporating prior information. Despite the Gaussian prior assumption, the approach adapts to cases where the posterior distribution may not be Gaussian but is absolutely continuous with respect to an appropriate Gaussian density. This adaptation is achieved through the Radon-Nikodym derivative, connecting the posterior distribution with the dominating Gaussian measure, often chosen as the prior. The algorithmic foundation of pCN lies in using stochastic processes that preserve either the posterior or prior distribution. These processes serve as proposals for Metropolis-Hastings methods with specific discretizations, ensuring preservation of the Gaussian reference measure.

### 2.5  Variational Inference in BNNs

The concept of variational inference has been applied in various forms to probabilistic models. The technique offers a way to approximate posterior distributions in Bayesian models (Jordan et al., 1999). The approximate distribution allows for a more feasible inference, especially for complex models like neural networks. In the context of BNNs, variational inference was brought into focus by Hinton & Van Camp (1993), which, while not explicitly termed as variational inference in the modern sense, laid the groundwork for later developments. A more direct application of variational inference to BNNs was detailed in the later works, such as those by Graves (2011) who applied variational inference to neural networks. Additionally, Kingma & Welling (2013) and Rezende et al. (2014) significantly contributed to popularizing and advancing the use of variational inference in deep learning and Bayesian neural networks through the introduction of efficient gradient-based optimization techniques.

---

**Algorithm 2** Variational Inference in Bayesian Neural Networks (BNNs)Blundell et al. (2015)

---

**Initialization:**
Choose an initial variational distribution $q_\theta(W)$ for the weights $W$ of BNN, parameterized by $\theta$.
Define the prior distribution $p(W)$ over the weights.

**while** not converged **do**
    **E-Step:** Estimate the Expectation of the log-likelihood over the variational distribution.
    Compute the gradient of the ELBO (Evidence Lower BOund) with respect to $\theta$, where

$$\text{ELBO}(\theta) = \mathbb{E}_{q_\theta(W)}[\log p(Y|X, W)] - \text{KL}[q_\theta(W)||p(W)]$$

    Here, $X$ and $Y$ are the inputs and outputs of the dataset, respectively, and KL denotes the Kullback-Leibler divergence between the variational distribution and the prior.

    **M-Step:** Maximize the ELBO with respect to $\theta$ by updating $\theta$ using gradient ascent:

$$\theta \leftarrow \theta + \eta\nabla_\theta\text{ELBO}(\theta)$$

    where $\eta$ is the learning rate.
**end while**

**Output:** Variational distribution $q_\theta(W)$ approximating the posterior distribution $p(W|X, Y)$.

---

This algorithm succinctly captures the iterative process of optimizing the parameters of a variational distribution to approximate the posterior distribution of a BNN's weights. Through the alternation of expectation (E-Step) and maximization (M-Step) phases, it seeks to minimize the difference between the variational distribution and the true posterior, leveraging the Evidence Lower Bound (ELBO) (Jordan et al., 1999) as a

tractable surrogate objective function. This approach enables the practical application of Bayesian inference to neural networks, facilitating the quantification of uncertainty in predictions and model parameters.

## 2.6 BNNs utilizing Laplace Approximation

Previous studies have shown that in the context of BNNs, the Laplace approximation serves as an efficient method for approximating the posterior distribution over the network's weights, striking a balance between the computational intensity of variational inference and the exhaustive nature of traditional sampling methods (Arbel et al., 2023; Blundell et al., 2015). This approximation is particularly appealing for its computational efficiency and its capability to facilitate theoretical analyses by transforming a complex posterior into a more tractable form. At the core of the Laplace approximation is the assumption that, around the loss function's minimum, the posterior distribution of the network's weights can be approximated by a Gaussian distribution. This is achieved by finding the mode of the posterior (equivalent to the minimum of the loss function in the Bayesian framework) and then approximating the curvature of the loss surface at this point using the Hessian matrix (Liang et al., 2018). The inverse of this Hessian is used to define the covariance of the Gaussian posterior, thus simplifying the representation of uncertainty in the model's predictions. This approach negates the need for direct optimization of the data likelihood with respect to the stochasticity of the network, which is a significant challenge with deep neural networks due to their complex loss landscapes (Daxberger et al., 2021).

MacKay (1992) was pioneering in extensively studying the Laplace approximation within BNNs, highlighting its utility in capturing predictive uncertainty, especially in data regions beyond the training set. Laplace Approximation in BNNs is a technique used to approximate the posterior distribution of the weights $W$ given data $X, Y$. This method approximates the posterior with a Gaussian distribution centered around the mode of the posterior, often referred to as the Maximum A Posteriori (MAP) estimate. The steps involved are as follows:

1. **Maximum A Posteriori (MAP) Estimation**: Find the MAP estimate of the weights $W_{\mathrm{MAP}}$ by optimizing the posterior distribution:

$$W_{\mathrm{MAP}} = \arg\min_W \left[ -\log p(Y|X, W) - \log p(W) \right]$$

where W's are the weights of BNN, parameterized by $\theta$. $p(Y|X, W)$ is the likelihood of observing the data given the weights, and $p(W)$ is the prior distribution over the weights.

2. **Gaussian Approximation of the Posterior**: The posterior $p(W|X, Y)$ is then approximated by a Gaussian distribution centered at $W_{\mathrm{MAP}}$ with covariance $\Sigma$ determined by the curvature of the log-posterior at $W_{\mathrm{MAP}}$:

$$q(W) \approx \mathcal{N}(W; W_{\mathrm{MAP}}, \Sigma)$$

The covariance matrix $\Sigma$ is the inverse of the Hessian of the negative log-posterior evaluated at $W_{\mathrm{MAP}}$:

$$\Sigma^{-1} = -\nabla\nabla \log p(W|X, Y)\big|_{W=W_{\mathrm{MAP}}}$$

3. **Predictive Distribution**: Predictions for new data $X^*$ are made by integrating over the approximate posterior:

$$p(Y^*|X^*, X, Y) \approx \int p(Y^*|X^*, W)q(W)dW$$

where $p(Y^*|X^*, X, Y)$ is the predictive distribution of the new data given the weights. Laplace Approximation offers a computationally efficient way to perform Bayesian inference in neural networks by providing a method to quantify uncertainty in model predictions through a Gaussian approximation of the posterior distribution. This approximation is particularly useful when the exact posterior is intractable or difficult to sample from directly.

## 2.7  BNNs utilizing MC-Dropout

Monte Carlo Dropout (Gal & Ghahramani, 2016) was introduced as a Bayesian approximation method to quantify model uncertainty in deep learning. The core idea behind this method is to interpret dropout, a technique commonly used to prevent overfitting in neural networks, from a Bayesian perspective. Normally, dropout randomly disables a fraction of neurons during the training phase to improve generalization. However, when viewed through the Bayesian lens, dropout can be seen as a practical way to approximate Bayesian inference in deep networks. This approximation allows the network to estimate not just a single set of weights, but a distribution over them, enabling the model to express uncertainty in its predictions. The MC Dropout technique involves running multiple forward passes through the network with dropout enabled. Each forward pass generates a different set of predictions due to the random omission of neurons, leading to a distribution of outputs for a given input.

## 2.8  Stochastic Weight Averaging-Gaussian

Stochastic Weight Averaging-Gaussian (SWAG) is a technique that builds upon the idea of Stochastic Weight Averaging (SWA) (Izmailov et al., 2018; Maddox et al., 2019). The core concept behind SWAG is to approximate the distribution of model weights by a Gaussian distribution, leveraging the empirical weight samples collected during training. This approach allows for a more nuanced understanding of the model's uncertainty compared to SWA, which simply averages weights over the latter part of the training process.

Mathematically, SWAG operates by first collecting a set of weights $\{W_i\}_{i=1}^N$ over the last $N$ epochs of training, where $W_i$ represents the weight vector at epoch $i$. The mean $\mu$ of the Gaussian distribution is computed as the simple average of these weights: $\mu = \frac{1}{N} \sum_{i=1}^N W_i$. To capture the covariance of the weight distribution, SWAG calculates the empirical covariance matrix using the formula $\Sigma = \frac{1}{N-1} \sum_{i=1}^N (W_i - \mu)(W_i - \mu)^T$. This formulation assumes a diagonal or low-rank plus diagonal approximation of the covariance matrix to maintain computational efficiency. The resulting Gaussian distribution, characterized by $\mu$ and $\Sigma$, can then be used for uncertainty estimation and prediction by sampling weights from this distribution and averaging the predictions of the resulting models.

# 3  Bayesian UQ for Neural Networks: Calibration-Emulation-Sampling

Traditional artificial neural networks (NN), such as feedforward and recurrent networks, typically consist of multiple layers. These networks are composed of an input layer, denoted as $l_0$, followed by a series of hidden layers $l_l$ for $l = 1, \ldots, m-1$, and concluding with an output layer $l_m$. In this architectural framework, comprising a total of $m+1$ layers, each layer $l$ is characterized by a linear transformation, which is subsequently subjected to a nonlinear operation $g$, commonly referred to as an activation function (Jospin et al., 2022):

$$l_0 = X, \tag{1}$$

$$l_l = g_l\left(W_l l_{l-1} + b_l\right) \quad \text{for all } l \in \{1, \cdots, m\}, \tag{2}$$

$$Y = l_m. \tag{3}$$

Here, $\theta = (W, b)$ are the parameters of the network, where $W$ are the weights of the network connections and $b$ the biases. A given NN architecture represents a set of functions isomorphic to the set of possible parameters $\theta$. Deep learning is the process of estimating the parameters $\theta$ from the training set $(X, Y) := \{(x_n, y_n)\}_{n=1}^N$, where training set is composed of a series of input $X$ and their corresponding labels $Y$ assuming $Y \in R^s$. Based on the training set, a neural network is trained to optimize network parameters $\theta$ in order to map $X \to Y$ with the objective of obtaining the maximal accuracy (under certain loss function $L(\cdot)$). Considering the error, we can write NN structure introduced in Equation 2 as a forward mapping, denoted as $\mathcal{G}$, that maps each parameter vector $\theta$ to a function that further connects $X$ to $Y$ with small errors $\varepsilon_n$ :

$$\mathcal{G} : \Theta \to Y^X, \quad \theta \mapsto \mathcal{G}(\theta) \tag{4}$$

More specifically,

$$\mathcal{G}(\boldsymbol{\theta}) : \boldsymbol{X} \to \boldsymbol{Y}, \quad \boldsymbol{y_n} = \hat{\boldsymbol{y}}_{\boldsymbol{n}} + \boldsymbol{\varepsilon_n}, \quad \hat{\boldsymbol{y}} = \mathcal{G}_n(\boldsymbol{X}; \theta) \tag{5}$$

where $\varepsilon$ represents random noise capturing disparity between the predicted and actual observed values in the training data. It is worth noting that the output $\boldsymbol{Y}$ could represent latent variables for classification problems.

To train NN, stochastic gradient algorithms could be used to solve the following optimization problem:

$$\boldsymbol{\theta}^* = \arg\min_{\boldsymbol{\theta} \in \Theta} L(\boldsymbol{\theta}; \boldsymbol{X}, \boldsymbol{Y}) = \arg\min_{\boldsymbol{\theta} \in \Theta} L(\boldsymbol{Y} - \mathcal{G}(\boldsymbol{X}; \theta)) \tag{6}$$

Note, we can define the log-likelihood based on the loss function $L(\boldsymbol{\theta}; \boldsymbol{X}, \boldsymbol{Y})$.

The point estimate approach, which is the traditional approach in deep learning, is relatively straightforward to implement with modern algorithms and software packages but tends to lack explainability (Yang et al., 2021). The resulting model might not be well calibrated (i.e., lack proper uncertainty quantification) (Guo et al., 2017; Nixon et al., 2019). Of all the techniques that exist to mitigate this, stochastic neural networks have proven to be one of the most generic and flexible solutions. Stochastic neural networks are a type of NN built by introducing stochastic components into the network. This is performed by giving the network either a stochastic activation or stochastic weights to simulate multiple possible models $\boldsymbol{\theta}$ with their associated probability distribution. The integration of stochastic components into neural networks allows for an extensive exploration of model uncertainty, which can be approached through Bayesian methods among others. It should be noted that not all neural networks that represent uncertainty are Bayesian or even stochastic; some employ deterministic methods to estimate uncertainty without relying on stochastic components or Bayesian inference. BNNs represent a subset of stochastic neural networks where Bayesian inference is specifically used for training, offering a rigorous probabilistic interpretation of model parameters. So a BNN can be defined as any stochastic artificial neural network trained using Bayesian inference (MacKay, 1992). The primary objective is to gain a deeper understanding of the uncertainty that underlies the processes the network is modeling.

To design a BNN, we put a prior distribution over the model parameters, $p(\boldsymbol{\theta})$, which leads to a prior confidence in the predictive power of the model $p(\boldsymbol{Y} \mid \boldsymbol{X}, \boldsymbol{\theta})$. By applying Bayes' theorem, and enforcing independence between the model parameters and the input, the Bayesian posterior can be written as:

$$\begin{aligned} p(\boldsymbol{\theta} \mid X, Y) &= \frac{p(Y \mid X, \boldsymbol{\theta}) \, p(\boldsymbol{\theta})}{\int_{\boldsymbol{\theta}} p(Y \mid X, \boldsymbol{\theta}') \, p(\boldsymbol{\theta}') \, d\boldsymbol{\theta}'} \\ &\propto p(Y \mid X, \boldsymbol{\theta}) \, p(\boldsymbol{\theta}). \end{aligned}$$

BNN is usually trained using MCMC algorithms. Because we typically have big amount of data, the likelihood evaluation tends to be expensive. One common approach to address this issue is subsampling, which restricts the computation to a subset of the data (see, for example, the stochastic gradient methods Hoffman et al. (2010); Welling & Teh (2011a); Chen et al. (2014)). The assumption is that there is redundancy in the data and an appropriate subset of the data can provide a good enough approximation of needed information of the full data set. In practice, it is a challenge to find good criteria and strategies for an effective subsampling in many applications. Additionally, subsampling could lead to a significant loss of accuracy Betancourt (2015).

Here, we propose a different approach that explores smoothness or regularity in parameter space. This characteristic in parameter space is true for most statistical models. Therefore, one would expect to find good and compact forms of approximation of functions (e.g., likelihood function) in parameter space. Sampling algorithms can use these approximate functions, also known as "surrogate" functions, to reduce their computational cost. More specifically, we propose using the CES scheme for high-dimensional BNN problems. Emulation bypasses the expensive evaluation of original forward models and reduces the cost of sampling to a small computational overhead. Compared with MCMC methods which require to repeatedly evaluate the original (large) neural network (NN) for the likelihood given the data, the proposed method builds a (smaller) NN emulator, which cuts the middle man (data) by mapping the parameters directly to the likelihood function to avoid its costly evaluation. That is, the emulator is trained based on the parameter-likelihood pairs,

which are collected through few iterations of the original NN. In contrast to subsampling methods, this approach can handle computationally intensive likelihood functions, whether the computational cost is due to high-dimensional data or complex likelihood function (e.g., models based on differential equations). Additionally, the calibration process increases the efficiency of MCMC algorithms by providing a robust initial point in the high-density region.

### 3.1 Calibration – Early stopping in Bayesian Neural Network

In the calibration step, our primary objective is to collect samples for model parameters to be used in the subsequent emulation. In the case of FBNN, we first set up a BNN model and sample from the model's posterior using Stochastic Gradient Hamilton Monte Carlo (SGHMC) algorithm. However, we do this only for a limited number of iterations to collect a small set of samples. These samples include both the model parameters ($\boldsymbol{\theta}^{(j)}$) and the outputs predicted by the model ($\mathcal{G}(\boldsymbol{X}; \boldsymbol{\theta}^{(j)})$) for each sample $j$ out of a total $J$ samples. The key focus of this training phase is not finding the target posterior distribution, but rather collecting a small number of posterior samples as the training data for the subsequent emulation step. Additionally, the last set of posterior samples obtained during calibration serves as the initial point for the Sampling phase in FBNN.

As mentioned earlier, to achieve our primary goal of collecting model parameters and the corresponding estimated outputs, we employ SGHMC algorithm in BNN. The SGHMC algorithm plays a crucial role in efficiently handling large datasets and collecting essential data during the calibration step of the FBNN. This algorithm is strategically chosen for the calibration step due to its effectiveness in exploring high-dimensional parameter spaces, especially when the sample size is also large. Its ability to introduce controlled stochasticity in updates proves instrumental in preventing local minima entrapment, thereby providing a comprehensive set of posterior samples that reflect the diversity of the parameter space.

During the calibration phase, we save samples of model parameters and their corresponding predictions at each iteration. The diverse set of samples obtained through SGHMC establishes a robust foundation for subsequent steps in the FBNN methodology. This strategic choice of SGHMC in the calibration step lays the groundwork for the emulation phase by contributing to the construction of a more adaptable emulator for the true neural network mapping. The broad coverage of the parameter space in the calibration step facilitates the generation of representative and diverse samples, further enhancing the overall efficiency and reliability of the FBNN methodology. In essence, the efficacy of SGHMC in exploring parameter spaces ensures a seamless transition from accurate parameter estimation to the construction of an adaptable emulator, making it a key component in the FBNN workflow.

The "calibration" step is instrumental in collecting a training set for the subsequent emulation step. It involves an early stopping strategy, aimed at collecting a targeted set of posterior samples without fully converging to the target distribution. The goal is to find an optimal collection of samples from the parameter space. This is aligned with traditional calibration goals of balancing accuracy and reliability, but within a new context.

### 3.2 Emulation – Deep Neural Network (DNN)

The original forward mapping in BNN involves mapping input dataset $X$ to response variable $Y$. For the likelihood evaluation using original forward mapping, it is necessary to calculate the likelihood $L(\boldsymbol{\theta}; X, Y)$ for each sample of model parameters. This means that with each iteration, when a new set of model parameters is introduced, the original forward mapping needs to be applied to generate output predictions, followed by the calculation of the likelihood. In general, this process can be very time-consuming. If, however, we have a small set of pairs of estimated model parameters and corresponding predicted outputs collected during the calibration step, an emulator can be trained on the recorded pairs. This essentially eliminates the intermediary step (passing through each data point), allowing us to map the parameters directly to the likelihood function through a simpler emulator, as opposed to plugging parameters into the original neural network for each parameter setting and evaluating the likelihood function. This leads to a computationally efficient likelihood evaluation. Based on this, to address the computational challenges of evaluating the likelihood with large datasets, an emulator $\mathcal{G}^e$ is constructed using the recorded pairs

$\{\boldsymbol{\theta}^{(j)}, \hat{\boldsymbol{y}}^{(j)} = \mathcal{G}(\boldsymbol{X}; \boldsymbol{\theta}^{(j)})\}_{j=1}^{J}$ obtained during the calibration step. In other words, these input-output pairs can be used to train a neural network model as an emulator $\mathcal{G}^e$ of the forward mapping $\mathcal{G}$:

$$\mathcal{G}^e(\boldsymbol{X}; \boldsymbol{\theta}) = DNN(\boldsymbol{\theta}, \mathcal{G}(\boldsymbol{X}; \boldsymbol{\theta})) \tag{7}$$

$$= F_{K-1} \circ \cdots \circ F_0(\boldsymbol{\theta}), \tag{8}$$

$$F_k(\cdot) = g_k(W_k \cdot + b_k) \in C\left(\mathbb{R}^{d_k}, \mathbb{R}^{d_{k+1}}\right) \tag{9}$$

Given a DNN model where $\boldsymbol{\theta}$ represents the input and $\mathcal{G}(\boldsymbol{X}; \boldsymbol{\theta})$ denotes the output, we set the dimensions as $d_0 = d$ and $d_K = s$. Here, the matrices $W_k$ are defined in the space $\mathbb{R}^{d_{k+1} \times d_k}$ and the vectors $b_k$ in $\mathbb{R}^{d_{k+1}}$. The functions $g_k$ act as (continuous) activation mechanisms.

There are various options for the activation functions, for instance,

$$g_k(x) = \left(\sigma\left(x_1\right), \ldots, \sigma\left(x_{d_{k+1}}\right)\right)$$

where $\sigma$ belongs to $C(\mathbb{R}, \mathbb{R})$ and includes the rectified linear unit (ReLU, $\sigma\left(x_i\right) = \max(0, x_i)$) and the leaky ReLU ($\sigma\left(x_i; \alpha\right) = x_i \cdot I\left(x_i \geq 0\right) + \alpha x_i \cdot I\left(x_i < 0\right)$). Alternatively,

$$g_k(x) = \left(\sigma_1(x), \ldots, \sigma_{d_{k+1}}(x)\right) \in C\left(\mathbb{R}^{d_{k+1}}, \mathbb{R}^{d_{k+1}}\right)$$

can be defined, where the $\{\sigma_i\}$ set is specified as the softmax function: $\sigma_i(x) = e^{x_i} / \sum_{j=1}^{d_{k+1}} e^{x_j}$. In the context of our numerical examples, the activation functions for the DNN are selected to ensure that both the function approximations and their derived gradients have minimized errors.

To elaborate, the process described involves optimizing the choice of activation functions within the DNN architecture to ensure that the network accurately approximates the target functions and their gradients. The method for selecting the optimal activation function is a grid search over a predefined set of activation functions to identify the suitable activation functions based on the best model performance

After the emulator is trained, the log-likelihood can be efficiently approximated as follows:

$$L(\boldsymbol{\theta}; X, Y) \approx L^e(\boldsymbol{\theta}; X, Y) = L(Y - \mathcal{G}^e(X; \boldsymbol{\theta})) \tag{10}$$

By combining the approximate likelihood $L^e(\boldsymbol{\theta}; X, Y)$ with the prior probability $p(\boldsymbol{\theta})$, an approximate posterior distribution $\boldsymbol{\theta} \mid X, Y$ that closely mirrors the true posterior distribution can be obtained.

Similarly, we could approximate the potential function using the predictions from DNN:

$$\Phi(\boldsymbol{\theta}^*) \approx \Phi^e(\boldsymbol{\theta}^*) = \Phi(\boldsymbol{\theta}; y) = \frac{1}{2}\|y - \mathcal{G}^e(X; \boldsymbol{\theta})\|_{\Gamma}^2 \tag{11}$$

Building upon the foundational concepts of using a DNN emulator $\mathcal{G}^e$ for approximating the forward mapping function $\mathcal{G}$, we further elaborate on the implications and advantages of this approach for Bayesian inference, particularly in the context of handling large datasets and/or complex likelihood functions. The emulation step, which involves training the DNN emulator with input-output pairs $\{\boldsymbol{\theta}^{(j)}, \mathcal{G}(\boldsymbol{X}; \boldsymbol{\theta}^{(j)})\}$, serves as a critical phase where the emulator learns to mimic the behavior of the original model with high accuracy. The utilization of DNN emulator to approximate the likelihood function in Bayesian inference presents a significant computational advantage over the direct use of the original BNN likelihood. This advantage stems primarily from the inherent differences in computational complexity between evaluating the the likelihood with a DNN emulator – which takes a set of model parameters as input and yields predicted responses—and the original BNN model – which processes $\boldsymbol{X}$ as input to produce the response variable.

In the sampling stage, the computational complexity could be significantly reduced if we use $\Phi^e$ instead of $\Phi$ in the accept/reject step of MCMC. If the emulator is a good representation of the forward mapping, the difference between $\Phi^e$ and $\Phi$ would be small and negligible. Then, the samples by such emulative MCMC have the stationary distribution with small discrepancy compared to the true posterior distribution. This approach not only ensures that the sampling process is computationally feasible but also maintains the integrity of the stationary distribution, closely approximating the true posterior distribution with minimal discrepancy. The integration of DNN emulators into the Bayesian inference workflow thus presents a compelling solution to the computational challenges associated with evaluating likelihood functions in complex models.

### 3.3 Sampling – Preconditioned Crank-Nicolson (pCN)

In the context of the FBNN method, the sampling step is crucial for exploring and exploiting the posterior distribution efficiently. The method employs MCMC algorithms based on a trained emulator to achieve full exploration and exploitation. However, challenges arise, especially in high-dimensional parameter spaces, where classical MCMC algorithms often exhibit escalating correlations among samples.

To address this issue, the pCN method introduced in algorithm1 has been used as a potential solution. Unlike classical methods, pCN avoids dimensional dependence challenges, making it particularly suitable for scenarios like BNN models with a high number of weights to be inferred (Hairer et al., 2009).

As explained in section 2.3, the pCN approach minimizes correlations between successive samples, a critical feature for ensuring the diversity and representativeness of the samples collected. This characteristic is vital for FBNNs, as it directly impacts the network's ability to learn from data and make robust predictions.

The emphasis on exploring the mode of the distribution is particularly relevant in high-dimensional spaces inherent to FBNN. The pCN method excels in traversing the parameter space with controlled perturbations, enhancing the algorithm's ability to capture the most probable configurations of model parameters. This focus on effective exploration around the mode contributes to a more accurate representation of the underlying neural network, ultimately improving model performance. In other words, the choice of pCN as the sampling method in FBNN is motivated by its tailored capacity to navigate and characterize the most probable regions of the parameter space. This choice reinforces the methodology's robustness and reliability, as pCN facilitates efficient sampling, leading to a more accurate and representative approximation of the posterior distribution.

Figure 1 displays a simulation that contrasts the sampling mechanisms of SGHMC and pCN within a multimodal probability distribution. The task is to sample from a mixture of 25 Gaussian distributions, represented in panel (a), using a total of 200,000 samples. Here, the target distribution is multimodal with several distinct peaks (modes). Middle figure shows that SGHMC has explored the parameter space, although with a less concentrated sampling around the modes compared to the target distribution. This indicates that while SGHMC is effective at exploring the space, it may not capture the modes as tightly as the target distribution. In the right figure related to pCN sampler, the concentration of samples around the modes is much higher compared to SGHMC, which indicates that pCN is more effective at exploring around the modes of the distribution. Our method combines these two algorithms to obtain the best of both worlds.

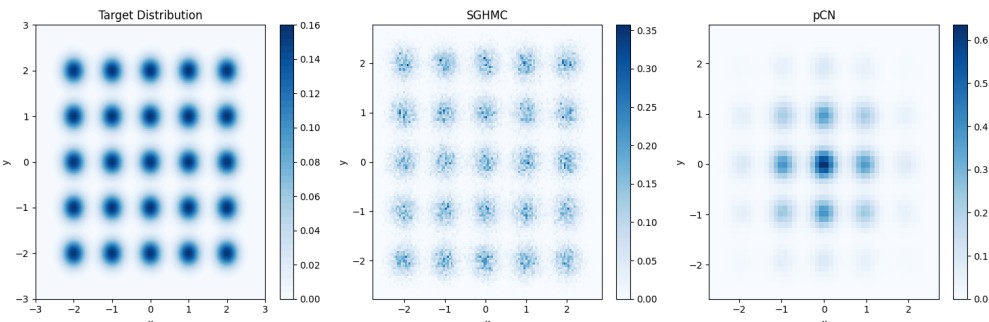

Figure 1: Sampling from a mixture of 25 Gaussians shown in (a) with 200k samples. SGHMC in (b) broadly explores the space, while pCN in (c) hones in on the high-density regions for precise mode capture.

### 3.4 Fast Bayesian Neural Network (FBNN).

Next, we combine all the techniques discussed above to speed up Bayesian UQ for BNN. More specifically, our main method called FBNN (Algorithm 3). This approach combines the strengths of BNN in uncertainty quantification, SGHMC for efficient parameter calibration, and the pCN method for sampling.

---

**Algorithm 3** Fast Bayesian Neural Network (FBNN)

---

**Input:** Training set $\{(X_n, Y_n)\}_{n=1}^N$, Prior $p(\boldsymbol{\theta})$
**Output:** Posterior samples for model parameters
**procedure** FBNN($\{(X_n, Y_n)\}_{n=1}^N, p(\boldsymbol{\theta})$)
    **Calibration Step:**
    Initialize model parameters $\boldsymbol{\theta}$ using SGHMC
    Save posterior samples $\{\boldsymbol{\theta}_n^{(j)}\}_{j=1}^J$ and the corresponding $\{\mathcal{G}_{\boldsymbol{\theta}_n^{(j)}}(\boldsymbol{X_n})\}_{j=1}^J$ after a few iterations
    **Emulation Step:**
    Build an emulator of the forward mapping $\mathcal{G}^e$ based on $\{\boldsymbol{\theta}_n^{(j)}, \mathcal{G}_{\boldsymbol{\theta}_n^{(j)}}(\boldsymbol{X_n})\}_{j=1}^J$ using a DNN as emulator.
    **Sampling Step:** Run approximate MCMC, particularly $pCN$, based on the emulator to propose $\theta'$
from $\theta$.
**end procedure**

---

## 4  Numerical Experiments

In this section, we present empirical evidence comparing our CES method with two baseline BNN methods equipped with the SGHMC and pCN samplers (shown as BNN-SGHMC and BNN-pCN). Additionally, we evaluate CES against various BNN architectures including those using Variational Inference, Lasso Approximation, MC-Dropout, SWAG, and RNS-HMC, all of which were detailed in Section 2. These comparisons are conducted through a series of regression and classification problems to thoroughly assess the performance and effectiveness of each method. We also include the results from DNN, which does not provide uncertainty quantification, but serves as a reference point. Moreover, we have incorporated the evaluation of Deep Ensembles, which consist of multiple DNNs each initialized with different random seeds, into our comparative analysis (shown as Ensemble-DNN). This comparative study emphasizes the strengths and weaknesses of each method, highlighting that, although the Ensemble-DNN approach allows for parallelization, it falls short in providing a probabilistic framework for analysis, a significant advantage offered by our CES method.

As discussed earlier, our main FBNN model utilizes SGHMC sampler in the calibration step and pCN in the sampling step. One of the distinctive features of our FBNN model lies in its strategic integration of the SGHMC sampler during the calibration step and the pCN algorithm during the sampling step. This combination is carefully chosen to harness the complementary strengths of these two optimization methods. Moreover, we extend our exploration by introducing three additional FBNN models: FBNN (pCN-SGHMC), where pCN is employed in the calibration step and SGHMC in the sampling step; FBNN (pCN-pCN), where pCN is used in both steps; and FBNN (SGHMC-SGHMC), where SGHMC is used in both calibration and sampling steps.

We evaluate these methods using a range of key metrics, including Mean Squared Error (MSE) for regression tasks and Accuracy for classification tasks. Additionally, we assess their performance in terms of computational cost, log probability evaluation, and various statistics related to the Effective Sample Size (ESS) of model parameters. These statistics include the minimum, maximum, and median ESS, as well as the minimum ESS per second. We also quantify the amount of speedup (denoted as "spdup"), a metric that compares the minimum ESS per second of the model with that of BNN-SGHMC as the benchmark.

Moreover, we evaluate the Coverage Probability (CP) for UQ in regression cases (CPs are set at 95%) and Expected Calibration Error (ECE) (Schefzik et al., 2013) and reliability diagrams (Bröcker & Smith, 2007) for UQ in classification cases. We also plot 95% Credible Interval (CI) constructed by the predicted outputs of the Bayesian models along with the average true output as a method for UQ in regression problems.

ECE specifically addresses model calibration, a component of UQ (Thorarinsdottir & Gneiting, 2010). A model that accurately estimates its uncertainty will not only perform well on average (accuracy, precision, recall) but also provide reliable probability estimates for its predictions. Models with lower ECE are considered to provide more reliable uncertainty estimates. Its predicted probabilities are more indicative of the true likelihood of outcomes, making it more trustworthy for decision-making under uncertainty.

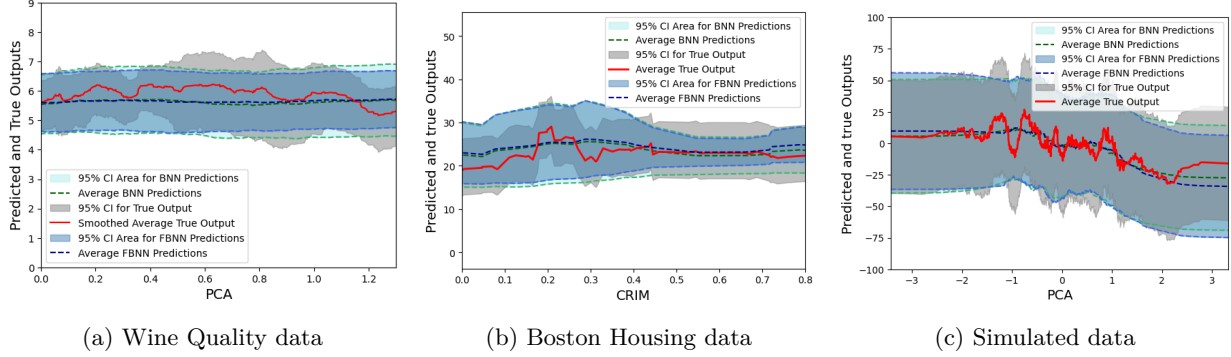

(a) Wine Quality data  (b) Boston Housing data  (c) Simulated data

Figure 2: Comparative Analysis of Predictive Credible Intervals and Mean Predictions for regression Data Sets. For each data set, the 95% CI for BNN predictions and FBNN predictions are shown as shaded areas. The average predictions from BNN and FBNN are represented with dashed lines. Additionally, the 95% CI for the true output as ground truth and the smoothed average true output are plotted as solid lines.

The reliability diagram is a common diagnostic graph used to summarize and evaluate probabilistic forecasts. They consist of a plot of the observed relative frequency against the predicted probability, providing a quick visual intercomparison when tuning probabilistic forecast systems, as well as documenting the performance of the final product. In these diagrams, the predicted probabilities are categorized into ten equally spaced segments, ranging from 0 to 1. Each forecast is placed into a bin corresponding to its predicted probability. Additionally, the diagram employs an equal frequency binning approach on the y-axis, ensuring a uniform distribution of data points across the observed frequency range.

Throughout these experiments, we collect 2000 posterior samples for the BNN-SGHMC and BNN-pCN. Samples have been collected at every iteration. In contrast, for the FBNN methods, we use a small number (200-400) samples from either BNN-SGHMC or BNN-pCN (depending on the specific FBNN model) along with the corresponding predicted outputs during the calibration step. These 200 samples serve as the training data for the emulator.

Additionally, for the BNN-SGHMC and BNN-pCN models, we train them based on a random initial starting point for the MCMC sampling. However, in the FBNN methods, we employ the set of posterior samples collected during the last iteration of the calibration step as the starting point for the subsequent MCMC sampling.

Including the performance data for the first 200 SGHMC base samples in Tables 2 and 1 reveals that these initial samples do not exhibit high quality when evaluated in terms of MSE for regression and accuracy for classification. The comparison indicates that, especially in regression cases, FBNN (SGHMC-pCN) significantly outperforms the 200 base SGHMC samples in terms of spdup across all cases examined. Furthermore, in classification datasets, FBNN (SGHMC-pCN) achieves better results in terms of Expected Calibration Error (ECE) compared to the 200 base SGHMC samples.

## 4.1 Regression

We first evaluate our proposed method using a set of simulate and real regression problems. The results are provided in Table 1.

### 4.1.1 Simulated Data

We begin our empirical evaluation by considering the following regression problem:

$$\boldsymbol{y} = \boldsymbol{\mathcal{G}_\theta}(\boldsymbol{X}) + \boldsymbol{\epsilon}, \quad \boldsymbol{\epsilon} \sim \boldsymbol{\mathcal{N}(0, \Gamma)} \tag{12}$$

The results are summarized in Table 1. For the simulated dataset, DNN provides the smallest MSE value at 0.29; however, BNN-SGHMC and FBNN (SGHMC-pCN) provide similar performance in terms of MSE.

Table 1: Performance of various deep learning methods based on regression problems. For ESS, minimum, median, and maximum values are provided in parenthesis.

| Dataset | Method | MSE | CP | Time (s) | ESS | minESS/s | spdup |
|---|---|---|---|---|---|---|---|
| Simulated | DNN | 0.29 | - | 39 | - | - | - |
| Dataset | Ensemble-DNN | 0.35 | 82.9% | 196 | - | - | - |
| | BNN-VI | 0.51 | 81.7% | 76 | - | - | - |
| | BNN-Lasso | 0.48 | 78.0% | 84 | - | - | - |
| | BNN-MC-Dropout | 2.34 | 86.2% | 32 | - | - | - |
| | BNN-SGHMC | 0.31 | 84.2% | 566 | (106, 831, 1522) | 0.19 | 1 |
| | BNN-SGHMC(first 200) | 2.34 | 90.3% | 52 | (12, 62, 147) | 0.22 | 1.15 |
| | SWAG | 0.45 | 82.2% | 150 | (96, 1021, 1461) | 0.64 | 3.36 |
| | BNN-RNS-HMC | 0.56 | 81.1% | 62 | (142, 973, 1476) | 2.29 | 12.05 |
| | BNN-pCN | 0.39 | 80.0% | 541 | (107, 844, 1533) | 0.20 | 1.05 |
| | FBNN (pCN-SGHMC) | 0.48 | 77.3% | 113 | (139, 1173, 1527) | 1.23 | 6.47 |
| | FBNN (pCN-pCN) | 0.38 | 71.2% | 115 | (106, 955, 1533) | 0.92 | 4.84 |
| | FBNN (SGHMC-SGHMC) | 0.43 | 75.6% | 62 | (139, 1173, 1527) | 2.35 | 12.36 |
| | FBNN (SGHMC-pCN) | 0.32 | 82.5% | 60 | (175, 821, 1536) | 2.93 | 15.55 |
| Wine | DNN | 0.43 | - | 26 | - | - | - |
| Quality | Ensemble-DNN | 0.41 | 47.4% | 137 | - | - | - |
| | BNN-VI | 0.66 | 39.5% | 28 | - | - | - |
| | BNN-Lasso | 0.63 | 40.9% | 42 | - | - | - |
| | BNN-MC-Dropout | 0.67 | 32.3% | 12 | - | - | - |
| | BNN-SGHMC | 0.53 | 51.3% | 505 | (111, 837, 1538) | 0.23 | 1 |
| | BNN-SGHMC(first 200) | 2.75 | 54.6% | 61 | (13, 111, 150) | 0.21 | 0.91 |
| | SWAG | 0.53 | 48.2% | 97 | (98, 1021, 1489) | 1.01 | 4.39 |
| | BNN-RNS-HMC | 0.62 | 44.7% | 38 | (107, 925, 1520) | 2.81 | 12.24 |
| | BNN-pCN | 0.65 | 51.1% | 620 | (99, 1003, 1532) | 0.16 | 0.69 |
| | FBNN (pCN-SGHMC) | 0.52 | 32.2% | 68 | (91, 912, 1533) | 1.37 | 5.95 |
| | FBNN (pCN-pCN) | 0.65 | 24.5% | 67 | (105, 1087, 1540) | 1.58 | 6.86 |
| | FBNN (SGHMC-SGHMC) | 0.50 | 40.0% | 70 | (77, 806, 1536) | 1.10 | 4.78 |
| | FBNN (SGHMC-pCN) | 0.52 | 48.1% | 57 | (92, 897, 1536) | 1.62 | 7.33 |
| Boston | DNN | 3.21 | - | 14 | - | - | - |
| Housing | Ensemble-DNN | 3.17 | 72.1% | 74 | - | - | - |
| | BNN-VI | 7.60 | 83.7% | 85 | - | - | - |
| | BNN-Lasso | 6.20 | 79.2% | 68 | - | - | - |
| | BNN-MC-Dropout | 10.12 | 81.2% | 91 | - | - | - |
| | BNN-SGHMC | 3.83 | 75.3% | 888 | (76, 649, 1536) | 0.09 | 1 |
| | BNN-SGHMC(first 200) | 7.40 | 66.9% | 86 | (9, 87, 150) | 0.09 | 1 |
| | SWAG | 5.81 | 71.9% | 104 | (68, 724, 1532) | 0.65 | 7.22 |
| | BNN-RNS-HMC | 9.42 | 73.4% | 76 | (73, 1032, 1504) | 0.96 | 10.6 |
| | BNN-pCN | 3.25 | 89.3% | 901 | (76, 649, 1536) | 0.08 | 0.88 |
| | FBNN (pCN-SGHMC) | 4.16 | 41.7% | 186 | (71, 965, 1543) | 0.38 | 4.22 |
| | FBNN (pCN-pCN) | 3.81 | 47.1% | 186 | (80, 966, 1541) | 0.43 | 4.78 |
| | FBNN (SGHMC-SGHMC) | 4.15 | 48.9% | 94 | (69, 979, 1542) | 0.74 | 8.22 |
| | FBNN (SGHMC-pCN) | 3.82 | 81.1% | 91 | (93, 938, 1543) | 1.03 | 11.94 |
| Alzheimer | DNN | 0.49 | - | 326 | - | - | - |
| Dataset | Ensemble-DNN | 0.42 | 89.3% | 1597 | - | - | - |
| | BNN-VI | 0.53 | 87.6% | 341 | - | - | - |
| | BNN-Lasso | 0.52 | 83.5% | 561 | - | - | - |
| | BNN-MC-Dropout | 0.60 | 92.8% | 268 | - | - | - |
| | BNN-SGHMC | 0.49 | 91.6% | 6524 | (102, 973, 1376) | 0.01 | 1 |
| | BNN-SGHMC(first 200) | 3.17 | 72.7% | 641 | (7, 82, 150) | 0.01 | 1 |
| | SWAG | 0.74 | 89.3% | 1214 | (106, 1002, 1542) | 0.08 | 8 |
| | BNN-RNS-HMC | 0.61 | 92.4% | 7324 | (96, 892, 1531) | 0.01 | 0.83 |
| | BNN-pCN | 0.51 | 89.9% | 6212 | (120, 1092, 1448) | 0.02 | 1.23 |
| | FBNN (pCN-SGHMC) | 0.50 | 90.2% | 643 | (116, 994, 1504) | 0.18 | 18 |
| | FBNN (pCN-pCN) | 0.56 | 91.4% | 632 | (108, 998, 1498) | 0.17 | 17 |
| | FBNN (SGHMC-SGHMC) | 0.55 | 88.4% | 671 | (118, 1012, 1541) | 0.17 | 17 |
| | FBNN (SGHMC-pCN) | 0.53 | 91.6% | 682 | (149, 984, 1497) | 0.22 | 22 |

Based on the results, BNN-MC-Dropout has the highest CP value (86.2%) compared to the other BNN and FBNN variants, indicating better coverage, but considerably higher MSE. This indicates that while BNN-MC-Dropout provides more reliable uncertainty estimates, its predictions are less accurate on average compared to those from other BNN and FBNN variants. Notably, among all the FBNN variants, FBNN (SGHMC-pCN) provides the highest CP at 82.5%, demonstrating a level of calibration comparable to that of the BNN model. The Ensemble-DNN demonstrates comparable performance to the FBNN (SGHMC-pCN) in terms of CP, yet it operates at a pace three times slower. As discussed earlier, the standard DNN does not quantify uncertainty.

Examining the efficiency of sample generation, all FBNN variants have relatively higher ESS per second values compared to all the other examined BNN models except BNN-RNS-HMC, and among all the models, FBNN (SGHMC-pCN) has the highest ESS per second at 2.93. This approach provides the highest speed-up

at 15.55 compared to BNN-SGHMC as the baseline model, highlighting its computational efficiency. While the speedup value of BNN-RNS-HMC is 12.05, which is higher than that of most FBNN models, it is still lower than our main FBNN model, FBNN (SGHMC-pCN), and its MSE is significantly higher.

Considering these results, FBNN (SGHMC-pCN) emerges as a strong performer, demonstrating a good balance between predictive accuracy and computational efficiency, making it a favorable choice for uncertainty quantification.

Figure 2c shows the estimated mean and prediction uncertainty for both BNN and FBNN (SGHMC-pCN) models, alongside the 95% CI for the true output as ground truth and the smoothed average true output. As we can see, BNN and FBNN have very similar and close credible intervals bounds. This consistency in credible interval bounds is significant for UQ, indicating that both models effectively and almost equally quantify uncertainty in their predictions. For clarity and conciseness within our figures, we have employed Principal Component Analysis (PCA) to transform the original feature dataset into a one-dimensional principal component. This one-dimensional principal component is then utilized as the x-axis for our plots in figure 2.

### 4.1.2 Wine Quality Data

As the first real dataset for the regression case, we use the Wine Quality data, provided by Cortez et al. (2009). This dataset contains various physicochemical properties of different wines, while the target variable is the quality rating.

Initially, we trained a DNN model with a total of $J = 241$ model parameters. Subsequently, we evaluated this model on a test dataset, recording both the model accuracy on the test dataset and the total training time.

For BNN models, we first place a prior distribution, $\mathcal{N}(\mathbf{0}, \mathbf{1}\boldsymbol{I})$, on the potential model parameters, denoted as p($\theta$). The BNN models also featured $J = 241$ total model parameters.

Based on the results shown in Table 1, DNN privides the best MSE at 0.43. However, as stated before, it lacks the capability to quantify uncertainty. Both BNN-SGHMC and FBNN (SGHMC-pCN) provide similar MSE values of 0.53 and 0.52 respectively. While BNN-SGHMC has slightly better CP, FBNN (SGHMC-pCN) is more computationally efficient. SWAG offers performance comparable to FBNN (SGHMC-pCN) in accuracy and CP, but it requires more computational resources. BNN-RNS-HMC stands out for its significant spdup among all the models, though it does not perform as well in terms of MSE and CP. Figure 2a shows the prediction mean and 95% CI for both methods, as well as 95% CI area for true output. For this dataset, we selected "Malic Acid" as the variable of interest for the x-axis.

### 4.1.3 Boston Housing Data

The Boston housing data was collected in 1978. Each of the 506 entries represent aggregated data of 14 features for homes from various suburbs in Boston. For this dataset, we followed CES steps similar to the Wine Quality dataset but with different number of model parameters ($J$=3009); the emulator has 3 hidden layers, but larger number of nodes (2048, 1024, 512).

For this example, the DNN achieves an MSE of 3.21. BNN-SGHMC and BNN-pCN exhibit comparable MSE values at 3.83 and 3.25, respectively, with BNN-pCN showing slightly better CP (89.3%) than BNN-SGHMC (85.3%). Among the FBNN variants, FBNN (SGHMC-pCN) stands out with a notable balance between MSE (3.82), CP (81.1%), and computational efficiency, completing the task in just 91 seconds. This model significantly outperforms all the other models in terms of speed-up (11.94), showcasing its effectiveness in sampling. Figure 2b shows the 95% CIs and mean predictions of both BNN-SGHMC and FBNN (SGHMC-pCN). In the case of the Boston housing dataset, "CRIM: per capita crime rate by town" serves as the x-axis variable.

Table 2: Performance of various deep learning methods based on classification problems.

| Dataset | Method | Acc | Time(s) | ESS(min,med,max) | minESS/s | spdup | ECE |
|---|---|---|---|---|---|---|---|
| Simulated | DNN | 96% | 18 | - | - | - | - |
| Dataset | Ensemble-DNN | 96% | 96 | - | - | - | 0.384 |
| | BNN-VI | 92% | 20 | - | - | - | 0.491 |
| | BNN-Lasso | 94% | 18 | - | - | - | 0.498 |
| | BNN-MC-Dropout | 92% | 16 | - | - | - | 0.326 |
| | BNN-SGHMC | 96% | 889 | (23, 189, 1468) | 0.03 | 1 | 0.450 |
| | BNN-SGHMC(first 200) | 81% | 86 | (7, 36, 134) | 0.08 | 2.66 | 0.498 |
| | SWAG | 83% | 98 | (67, 742, 1503) | 0.68 | 22.78 | 0.460 |
| | BNN-RNS-HMC | 72% | 27 | (137, 1024, 1523) | 5.07 | 169 | 0.492 |
| | BNN-pCN | 96% | 1015 | (38, 176, 1351) | 0.04 | 1.46 | 0.455 |
| | FBNN (pCN-SGHMC) | 92% | 105 | (132, 947, 1539) | 1.26 | 48.92 | 0.390 |
| | FBNN (pCN-pCN) | 90% | 105 | (147, 861, 1535) | 1.41 | 54.44 | 0.405 |
| | FBNN (SGHMC-SGHMC) | 95% | 97 | (153, 871, 1520) | 1.57 | 61.04 | 0.389 |
| | FBNN (SGHMC-pCN) | 96% | 92 | (153, 871, 1520) | 1.67 | 64.32 | 0.380 |
| Adult | DNN | 85% | 426 | - | - | - | - |
| Dataset | Ensemble-DNN | 84% | 2153 | - | - | - | 0.556 |
| | BNN-VI | 80% | 562 | - | - | - | 0.642 |
| | BNN-Lasso | 83% | 256 | - | - | - | 0.631 |
| | BNN-MC-Dropout | 82% | 187 | - | - | - | 0.540 |
| | BNN-SGHMC | 83% | 5979 | (16, 202, 1520) | 0.002 | 1 | 0.574 |
| | BNN-SGHMC(first 200) | 78% | 581 | (1.13, 41.44, 148.43) | 0.002 | 0.95 | 0.594 |
| | SWAG | 79% | 1641 | (47, 912, 1532) | 0.04 | 14.32 | 0.668 |
| | BNN-RNS-HMC | 72% | 6110 | (89, 960, 1530) | 0.01 | 5 | 0.658 |
| | BNN-pCN | 83% | 6227 | (9, 117, 1518) | 0.001 | 0.52 | 0.616 |
| | FBNN (pCN-SGHMC) | 82% | 642 | (87, 892, 1539) | 0.13 | 50.93 | 0.580 |
| | FBNN (pCN-pCN) | 82% | 639 | (88, 890, 1540) | 0.12 | 51.52 | 0.592 |
| | FBNN (SGHMC-SGHMC) | 83% | 612 | (68, 941, 1541) | 0.12 | 41.88 | 0.583 |
| | FBNN (SGHMC-pCN) | 84% | 609 | (89, 875, 1539) | 0.15 | 54.91 | 0.576 |
| Alzheimer | DNN | 82% | 51 | - | - | - | - |
| Dataset | Ensemble-DNN | 83% | 262 | - | - | - | 0.542 |
| | BNN-VI | 72% | 61 | - | - | - | 0.546 |
| | BNN-Lasso | 76% | 256 | - | - | - | 0.524 |
| | BNN-MC-Dropout | 76% | 12 | - | - | - | 0.429 |
| | BNN-SGHMC | 81% | 2736 | (81, 588, 1526) | 0.03 | 1 | 0.499 |
| | BNN-SGHMC(first 200) | 69% | 282 | (8, 84, 149) | 0.03 | 0.96 | 0.523 |
| | SWAG | 81% | 312 | (72, 913, 1562) | 0.23 | 7.69 | 0.508 |
| | BNN-RNS-HMC | 58% | 293 | (84, 915, 1540) | 0.28 | 9.55 | 0.521 |
| | BNN-pCN | 73% | 2660 | (71, 424, 1534) | 0.03 | 1 | 0.469 |
| | FBNN (pCN-SGHMC) | 76% | 277 | (76, 947, 1542) | 0.27 | 9 | 0.568 |
| | FBNN (pCN-pCN) | 77% | 274 | (70, 931, 1542) | 0.25 | 8.33 | 0.377 |
| | FBNN (SGHMC-SGHMC) | 80% | 278 | (81, 973, 1538) | 0.28 | 9.33 | 0.448 |
| | FBNN (SGHMC-pCN) | 84% | 280 | (92, 914, 1535) | 0.33 | 11 | 0.376 |

### 4.1.4 Alzheimer Data

We have expanded our experimental evaluation to include a dataset from the National Alzheimer's Coordinating Center(NACC). NACC is responsible for developing and maintaining a database of patient information collected from the 29 Alzheimer disease centers (ADCs) funded by the National Institute on Aging (NIA) (Beekly et al., 2004). The NIA appointed the ADC Clinical Task Force to determine and define an expanded, standardized clinical data set, called the Uniform Data Set (UDS). The goal of the UDS is to provide ADC researchers a standard set of assessment procedures to better characterize ADC participants with mild Alzheimer disease and mild cognitive impairment in comparison with nondemented controls (Beekly et al., 2007).This dataset includes records from 185,831 subjects, with a rich array of 1,024 features, narrowed down to 56 key features for our analysis. These features were carefully selected to represent a wide spectrum of variables relevant to Alzheimer's Disease (AD) diagnosis, including functional abilities, brain morphometrics, living situations, and demographic information (Ren et al., 2022). For the regression case, the goal is to predict Left Hippocampus Volume, a critical marker in the progression of AD, as a function of other variables.

For this dataset, the DNN model achieves an MSE of 0.49, serving as a baseline for the performance of more complex models. BNNs using different approaches (Variational Inference, Lasso, MC-Dropout, Stochastic Gradient Hamiltonian Monte Carlo (SGHMC)) show a range of MSE values from 0.49 to 0.60. Notably, BNN-MC-Dropout stands out with the highest CP of 92.8%, though its MSE is high at 0.60. Among the FBNN models, the FBNN (SGHMC-pCN) model is highlighted for its balanced performance with an MSE of 0.53 and a CP of 91.6% which is close to highest CP among all different models (92.8% for BNN-MC-Dropout

Figure 3: Reliability Diagrams for Simulated dataset in classification task. These diagrams incorporate equal frequency binning.

). It shows a considerable improvement in computational efficiency, evidenced by a computational time of 682 seconds and a spdup factor of 22 times compared to BNN-SGHMC as baseline BNN model.

### 4.2   Classification

We also evaluate our method based on a set of simulated and real classification problems. The results are provided in Table 2.

#### 4.2.1   Simulated Data

In this section we focus on a binary classification problem using simulate data. The data setup involves a similar structure to the regression case, with the key difference being in the response variable, loss functions, and the architecture of the DNN emulator. Here, the emulator has two layers with ReLU activation functions for the hidden layers and sigmoid activation for the output layer. FBNN (SGHMC-pCN), DNN, Ensemble-DNN, BNN-SGHMC, and BNN-pCN exhibit comparable performance in terms of accuracy and outperform other models. Conversely, BNN-RNS-HMC and SWAG demonstrate considerably lower accuracy relative to the aforementioned methods. While BNN-RNS-HMC achieves the highest spdup, it significantly underperforms in terms of accuracy. In contrast, FBNN (SGHMC-pCN) boasts the second-highest speedup, at 64.32, showcasing its computational efficiency relative to BNN-SGHMC. Furthermore, it maintains the highest accuracy rate of 96%, indicating an optimal balance between computational efficiency and accuracy.

BNN-Lasso, BNN-RNS-HMC and BNN-VI have relatively high ECE values, suggesting less reliable UQ compared to other methods. BNN-MC-Dropout has the lowest ECE value compared to other methods, but it also have a lower accuracy rate. Among the FBNN variants, FBNN (SGHMC-pCN) presents a low ECE, closely aligning with the ECE value of BNN-MC-Dropout, while providing an accuracy rate similar to DNN.

Moreover, in terms of UQ, a model with a reliability curve that closely follows the diagonal line is considered better calibrated, meaning its predicted probabilities are more aligned with actual outcomes. The FBNN variants, particularly the first FBNN (SGHMC-pCN) plot, appear to be better calibrated across most probability ranges except at the highest probabilities, suggesting a more reliable UQ. However, it is important to note that while the models exhibit overconfidence at the high end of predicted values, this may also be due to fewer data points in those bins, which is a common issue in reliability diagrams.

#### 4.2.2   Adult Data

Next, we use the Adult dataset, discussed in Becker & Kohavi (1996). This dataset comprises a relatively larger sample size of 40,434 data points, each described by 14 distinct features. All BNN and FBNN versions contain a complex structure with 2,761 internal model parameters. The classification task for the Adult dataset involves predicting whether an individual will earn more or less than \$50,000 per year.

All methods achieve comparable accuracy rates, though BNN-RNS-HMC and SWAG exhibit slightly lower accuracy in comparison to the others. Among them, FBNN (SGHMC-pCN) stands out as the most computationally efficient method for uncertainty quantification, boasting a spdup value of 54.91 relative to the baseline BNN approach. A low ECE value of the FBNN (SGHMC-pCN) model, along with a reliability

curve that closely aligns with the diagonal line (not depicted in the paper), signifies the model's superior performance in terms of uncertainty quantification.

### 4.2.3   Alzheimer Data

For the classification case, we utilized the same NACC dataset previously introduced in the regression context. Here, our objective is to predict cognitive status, classifying individuals as either cognitively unimpaired (healthy controls, HC), labeled as class 0, or as having mild cognitive impairment (MCI) due to Alzheimer's disease (AD) or dementia due to AD, labeled as class 1. Still we use records from 185,831 subjects with 56 key features for our analysis.

The DNN and Ensemble-DNN models achieve an accuracy of 82% and 83%, setting a baseline for comparison with the Bayesian approaches. Various BNN methods show a range of accuracies, with BNN-SGHMC achieving a notable 81% accuracy but requiring a significantly higher computational cost (2736 seconds). The BNN-VI and BNN-pCN models show lower accuracies of 72% and 73%, respectively, with BNN-pCN also having a high computational demand (2660 seconds). With an accuracy of 84% at a relatively low computational cost (280 seconds), FBNN (SGHMC-pCN) surpasses all other models in correctly identifying Alzheimer's disease, making it the most reliable model among those tested. This model not only surpasses the accuracy of the standard DNN and Ensemble-DNN models but also offers a balance between computational efficiency and high accuracy. Moreover, FBNN (SGHMC-pCN) model demonstrates the highest sampling efficiency (spdup of 11), indicating it can achieve high accuracy with fewer computational resources compared to other models. Moreover, for the FBNN model implemented on the Alzheimer dataset, the ECE value is low, and the reliability curve closely tracks the diagonal line, indicating the model's adeptness at UQ for this specific dataset.

## 5   Conclusion

In this paper, we have proposed an innovative CES framework called FBNN, specifically designed to enhance the computational efficiency and scalability of BNN for high-dimensional data. Our primary goal is to provide a robust solution for uncertainty quantification in high-dimensional spaces, leveraging the power of Bayesian principles while mitigating the computational bottlenecks traditionally associated with BNN.

In our numerical experiments, we have successfully applied various variants of FBNN, including different configurations with BNN, to regression and classification tasks on both synthetic and real datasets. Remarkably, the FBNN variant incorporating SGHMC for calibration and pCN for sampling, denoted as FBNN (SGHMC-pCN), not only matches the predictive accuracy of traditional BNN but also offers substantial computational advantages. The superior performance of the FBNN variant employing SGHMC for calibration and pCN for sampling can be attributed to the complementary strengths of these two samplers. SGHMC excels at broad exploration of the parameter space, providing an effective means for understanding the global structure during the calibration step. On the other hand, pCN is adept at efficient sampling around modes, offering a valuable tool for capturing local intricacies in the distribution during the final sampling step. By combining these samplers in the FBNN model, we achieve a balanced approach between exploration (calibration with SGHMC) and exploitation (final sampling with pCN).

Future work could involve extending our method to more complex problems (e.g., spatiotemporal data) and complex network structures (e.g., graph neural networks). Additionally, future research could focus on improving the emulation step by optimizing the DNN architecture. Finally, our method could be further improved by embedding the sampling algorithm in an adaptive framework similar to the method of Zhang et al. (2018).

## References

S. Ahn, B. Shahbaba, and M. Welling. Distributed Stochastic Gradient MCMC. In International Conference on Machine Learning, 2014.

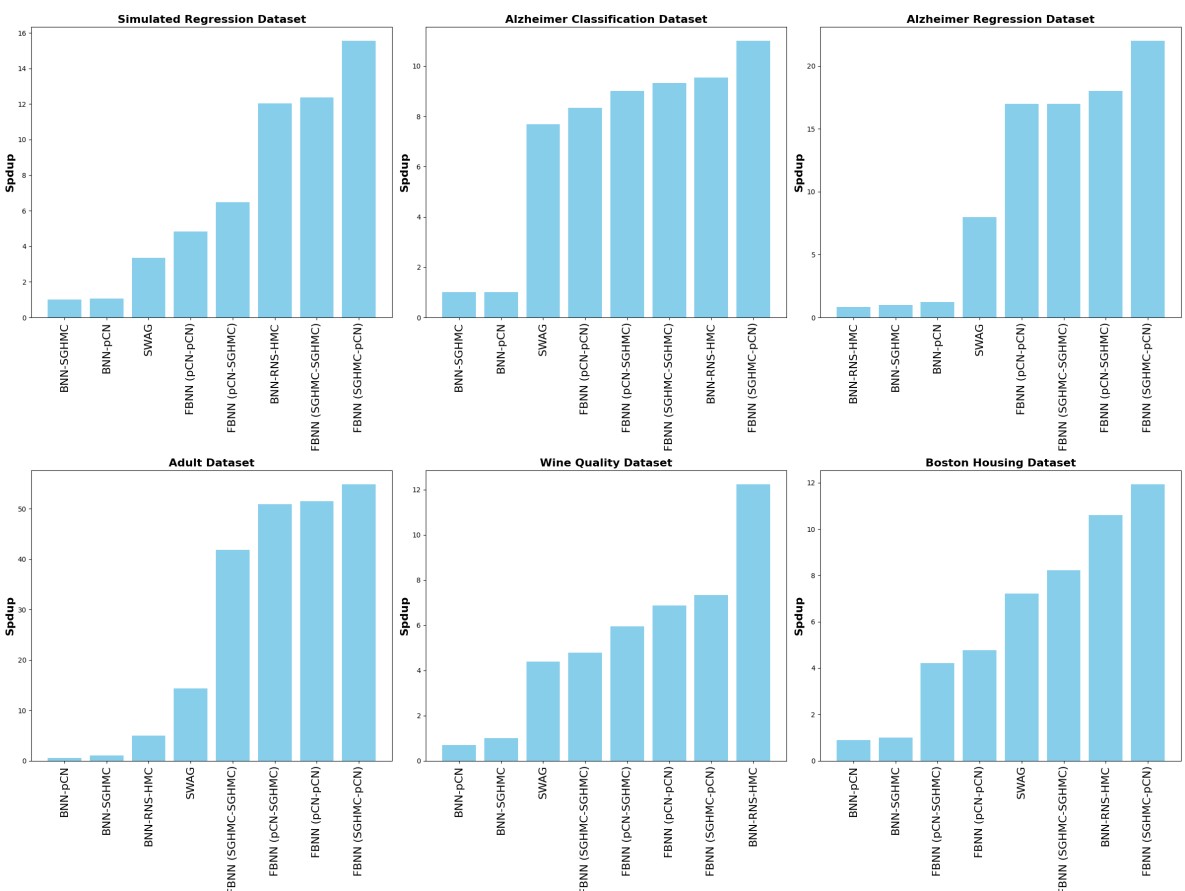

Figure 4: Comparative analysis of spdup for various BNN methods across all tested datasets. Methods are ordered by efficiency within each dataset, highlighting the impact of model characteristics on sampling performance.

Christophe Andrieu, Nando De Freitas, Arnaud Doucet, and Michael I Jordan. An introduction to mcmc for machine learning. Machine learning, 50:5–43, 2003.

Julyan Arbel, Konstantinos Pitas, Mariia Vladimirova, and Vincent Fortuin. A primer on bayesian neural networks: review and debates. arXiv preprint arXiv:2309.16314, 2023.

Barry Becker and Ronny Kohavi. Adult. UCI Machine Learning Repository, 1996.

Duane L Beekly, Erin M Ramos, Gerald van Belle, Woodrow Deitrich, Amber D Clark, Mary E Jacka, Walter A Kukull, et al. The national alzheimer's coordinating center (nacc) database: an alzheimer disease database. Alzheimer Disease & Associated Disorders, 18(4):270–277, 2004.

Duane L Beekly, Erin M Ramos, William W Lee, Woodrow D Deitrich, Mary E Jacka, Joylee Wu, Janene L Hubbard, Thomas D Koepsell, John C Morris, Walter A Kukull, et al. The national alzheimer's coordinating center (nacc) database: the uniform data set. Alzheimer Disease & Associated Disorders, 21(3): 249–258, 2007.

Alexandros Beskos. A stable manifold mcmc method for high dimensions. Statistics & Probability Letters, 90:46–52, 2014.

Alexandros Beskos, Gareth Roberts, and Andrew Stuart. Optimal scalings for local Metropolis–Hastings chains on nonproduct targets in high dimensions. The Annals of Applied Probability, 19(3):863 – 898, 2009. doi: 10.1214/08-AAP563. URL https://doi.org/10.1214/08-AAP563.

Alexandros Beskos, Frank J Pinski, Jesús Marıa Sanz-Serna, and Andrew M Stuart. Hybrid monte carlo on hilbert spaces. Stochastic Processes and their Applications, 121(10):2201–2230, 2011.

Alexandros Beskos, Mark Girolami, Shiwei Lan, Patrick E Farrell, and Andrew M Stuart. Geometric mcmc for infinite-dimensional inverse problems. Journal of Computational Physics, 335:327–351, 2017.

Michael Betancourt. The fundamental incompatibility of scalable hamiltonian monte carlo and naive data subsampling. In International Conference on Machine Learning, pp. 533–540. PMLR, 2015.

Charles Blundell, Julien Cornebise, Koray Kavukcuoglu, and Daan Wierstra. Weight uncertainty in neural network. In International conference on machine learning, pp. 1613–1622. PMLR, 2015.

Edwin V Bonilla, Kian Chai, and Christopher Williams. Multi-task gaussian process prediction. Advances in neural information processing systems, 20, 2007.

Jochen Bröcker and Leonard A Smith. Increasing the reliability of reliability diagrams. Weather and forecasting, 22(3):651–661, 2007.

Tianqi Chen, Emily Fox, and Carlos Guestrin. Stochastic gradient hamiltonian monte carlo. In International conference on machine learning, pp. 1683–1691. PMLR, 2014.

Jian Cheng, Pei-song Wang, Gang Li, Qing-hao Hu, and Han-qing Lu. Recent advances in efficient computation of deep convolutional neural networks. Frontiers of Information Technology & Electronic Engineering, 19:64–77, 2018.

Siddhartha Chib and Edward Greenberg. Understanding the metropolis-hastings algorithm. The american statistician, 49(4):327–335, 1995.

Emmet Cleary, Alfredo Garbuno-Inigo, Shiwei Lan, Tapio Schneider, and Andrew M. Stuart. Calibrate, emulate, sample. Journal of Computational Physics, 424:109716, jan 2021. doi: 10.1016/j.jcp.2020.109716. URL https://doi.org/10.1016%2Fj.jcp.2020.109716.

Paulo Cortez, A. Cerdeira, F. Almeida, T. Matos, and J. Reis. Wine Quality. UCI Machine Learning Repository, 2009. DOI: https://doi.org/10.24432/C56S3T.

S. L. Cotter, G. O. Roberts, A. M. Stuart, and D. White. MCMC Methods for Functions: Modifying Old Algorithms to Make Them Faster. Statistical Science, 28(3):424 – 446, 2013. doi: 10.1214/13-STS421. URL https://doi.org/10.1214/13-STS421.

Tiangang Cui, Kody JH Law, and Youssef M Marzouk. Dimension-independent likelihood-informed mcmc. Journal of Computational Physics, 304:109–137, 2016.

Carla Currin, Toby Mitchell, Max Morris, and Don Ylvisaker. A bayesian approach to the design and analysis of computer experiments. Technical report, Oak Ridge National Lab., TN (USA), 1988.

Giuseppe Da Prato and Jerzy Zabczyk. Stochastic equations in infinite dimensions. Cambridge university press, 2014.

Shaveta Dargan, Munish Kumar, Maruthi Rohit Ayyagari, and Gulshan Kumar. A survey of deep learning and its applications: a new paradigm to machine learning. Archives of Computational Methods in Engineering, 27:1071–1092, 2020.

Erik Daxberger, Agustinus Kristiadi, Alexander Immer, Runa Eschenhagen, Matthias Bauer, and Philipp Hennig. Laplace redux-effortless bayesian deep learning. Advances in Neural Information Processing Systems, 34:20089–20103, 2021.

Yarin Gal and Zoubin Ghahramani. Dropout as a bayesian approximation: Representing model uncertainty in deep learning. In international conference on machine learning, pp. 1050–1059. PMLR, 2016.

Jacob Gardner, Geoff Pleiss, Kilian Q Weinberger, David Bindel, and Andrew G Wilson. Gpytorch: Blackbox matrix-matrix gaussian process inference with gpu acceleration. Advances in neural information processing systems, 31, 2018.

Andrew Gelman, Walter R Gilks, and Gareth O Roberts. Weak convergence and optimal scaling of random walk metropolis algorithms. The annals of applied probability, 7(1):110–120, 1997.

Alex Graves. Practical variational inference for neural networks. Advances in neural information processing systems, 24, 2011.

Chuan Guo, Geoff Pleiss, Yu Sun, and Kilian Q. Weinberger. On calibration of modern neural networks. In Doina Precup and Yee Whye Teh (eds.), Proceedings of the 34th International Conference on Machine Learning, volume 70 of Proceedings of Machine Learning Research, pp. 1321–1330. PMLR, 06–11 Aug 2017. URL https://proceedings.mlr.press/v70/guo17a.html.

Martin Hairer, Andrew M Stuart, and Jochen Voss. Sampling conditioned diffusions. Trends in stochastic analysis, 353:159–186, 2009.

Dave Higdon, Marc Kennedy, James C. Cavendish, John A. Cafeo, and Robert D. Ryne. Combining field data and computer simulations for calibration and prediction. SIAM Journal on Scientific Computing, 26(2): 448–466, 2004. doi: 10.1137/S1064827503426693. URL https://doi.org/10.1137/S1064827503426693.

Geoffrey E Hinton and Drew Van Camp. Keeping the neural networks simple by minimizing the description length of the weights. In Proceedings of the sixth annual conference on Computational learning theory, pp. 5–13, 1993.

M. Hoffman and A. Gelman. The No-U-Turn Sampler: Adaptively Setting Path Lengths in Hamiltonian Monte Carlo. arxiv.org/abs/1111.4246, 2011.

Matthew D. Hoffman, David M. Blei, and Francis R. Bach. Online learning for latent dirichlet allocation. In John D. Lafferty, Christopher K. I. Williams, John Shawe-Taylor, Richard S. Zemel, and Aron Culotta (eds.), NIPS, pp. 856–864. Curran Associates, Inc., 2010.

Pavel Izmailov, Dmitrii Podoprikhin, Timur Garipov, Dmitry Vetrov, and Andrew Gordon Wilson. Averaging weights leads to wider optima and better generalization. arXiv preprint arXiv:1803.05407, 2018.

Tommi S Jaakkola and Michael I Jordan. Bayesian parameter estimation via variational methods. Statistics and Computing, 10:25–37, 2000.

Michael I Jordan, Zoubin Ghahramani, Tommi S Jaakkola, and Lawrence K Saul. An introduction to variational methods for graphical models. Machine learning, 37:183–233, 1999.

Laurent Valentin Jospin, Hamid Laga, Farid Boussaid, Wray Buntine, and Mohammed Bennamoun. Hands-on bayesian neural networks—a tutorial for deep learning users. IEEE Computational Intelligence Magazine, 17(2):29–48, 2022.

Marc C. Kennedy and Anthony O'Hagan. Bayesian Calibration of Computer Models. Journal of the Royal Statistical Society Series B: Statistical Methodology, 63(3):425–464, 01 2002. ISSN 1369-7412. doi: 10. 1111/1467-9868.00294. URL https://doi.org/10.1111/1467-9868.00294.

Diederik P Kingma and Max Welling. Auto-encoding variational bayes. arXiv preprint arXiv:1312.6114, 2013.

Yongchan Kwon, Joong-Ho Won, Beom Joon Kim, and Myunghee Cho Paik. Uncertainty quantification using bayesian neural networks in classification: Application to ischemic stroke lesion segmentation. In Medical Imaging with Deep Learning, 2022.

Balaji Lakshminarayanan, Alexander Pritzel, and Charles Blundell. Simple and scalable predictive uncertainty estimation using deep ensembles. Advances in neural information processing systems, 30, 2017.

S. Lan, J. A. Palacios, M. Karcher, V. Minin, and B Shahbaba. An efficient bayesian inference framework for coalescent-based nonparametric phylodynamics. Bioinformatics, 31(20):3282–3289, 2015.

Shiwei Lan, Shuyi Li, and Babak Shahbaba. Scaling up bayesian uncertainty quantification for inverse problems using deep neural networks. SIAM/ASA Journal on Uncertainty Quantification, 10(4):1684–1713, 2022a. doi: 10.1137/21M1439456.

Shiwei Lan, Shuyi Li, and Babak Shahbaba. Scaling up bayesian uncertainty quantification for inverse problems using deep neural networks, 2022b.

Kody JH Law. Proposals which speed up function-space mcmc. Journal of Computational and Applied Mathematics, 262:127–138, 2014.

Yann LeCun, Léon Bottou, Yoshua Bengio, and Patrick Haffner. Gradient-based learning applied to document recognition. Proceedings of the IEEE, 86(11):2278–2324, 1998.

Lingge Li, Andrew Holbrook, Babak Shahbaba, and Pierre Baldi. Neural Network Gradient Hamiltonian Monte Carlo. Computational Statistics, 34(1):281–299, 2019a.

Lingge Li, Andrew Holbrook, Babak Shahbaba, and Pierre Baldi. Neural network gradient hamiltonian monte carlo. Computational statistics, 34:281–299, 2019b.

Faming Liang, Qizhai Li, and Lei Zhou. Bayesian neural networks for selection of drug sensitive genes. Journal of the American Statistical Association, 113(523):955–972, 2018.

Haitao Liu, Yew-Soon Ong, Xiaobo Shen, and Jianfei Cai. When gaussian process meets big data: A review of scalable gps. IEEE transactions on neural networks and learning systems, 31(11):4405–4423, 2020.

David JC MacKay. A practical bayesian framework for backpropagation networks. Neural computation, 4 (3):448–472, 1992.

Wesley J Maddox, Pavel Izmailov, Timur Garipov, Dmitry P Vetrov, and Andrew Gordon Wilson. A simple baseline for bayesian uncertainty in deep learning. Advances in neural information processing systems, 32, 2019.

Nicholas Metropolis, Arianna W Rosenbluth, Marshall N Rosenbluth, Augusta H Teller, and Edward Teller. Equation of state calculations by fast computing machines. The journal of chemical physics, 21(6):1087–1092, 1953.

Radford M Neal. Bayesian learning for neural networks, volume 118. Springer Science & Business Media, 2012.

Radford M Neal et al. Mcmc using hamiltonian dynamics. Handbook of markov chain monte carlo, 2(11): 2, 2011.

Jeremy Nixon, Michael W. Dusenberry, Linchuan Zhang, Ghassen Jerfel, and Dustin Tran. Measuring calibration in deep learning. In Proceedings of the IEEE/CVF Conference on Computer Vision and Pattern Recognition (CVPR) Workshops, June 2019.

Jeremy Oakley and Anthony O'Hagan. Bayesian inference for the uncertainty distribution of computer model outputs. Biometrika, 89(4):769–784, 12 2002. ISSN 0006-3444. doi: 10.1093/biomet/89.4.769. URL https://doi.org/10.1093/biomet/89.4.769.

Jeremy E. Oakley and Anthony O'Hagan. Probabilistic sensitivity analysis of complex models: A bayesian approach. Journal of the Royal Statistical Society. Series B (Statistical Methodology), 66(3):751–769, 2004. ISSN 13697412, 14679868. URL http://www.jstor.org/stable/3647504.

A. O'Hagan. Bayesian analysis of computer code outputs: A tutorial. Reliability Engineering & System Safety, 91(10):1290–1300, 2006. ISSN 0951-8320. doi: https://doi.org/10.1016/j.ress.2005.11.025. URL https://www.sciencedirect.com/science/article/pii/S0951832005002383. The Fourth International Conference on Sensitivity Analysis of Model Output (SAMO 2004).

Michael P Perrone and Leon N Cooper. When networks disagree: Ensemble methods for hybrid neural networks. In How We Learn; How We Remember: Toward An Understanding Of Brain And Neural Systems: Selected Papers of Leon N Cooper, pp. 342–358. World Scientific, 1995.

Yueqi Ren, Babak Shahbaba, and Craig E Stark. Hierarchical, multiclass prediction of alzheimer's clinical diagnosis using imputed, multimodal nacc data. Alzheimer's & Dementia, 18:e066698, 2022.

Danilo Jimenez Rezende, Shakir Mohamed, and Daan Wierstra. Stochastic backpropagation and approximate inference in deep generative models. In International conference on machine learning, pp. 1278–1286. PMLR, 2014.

Christian P Robert, George Casella, and George Casella. Monte Carlo statistical methods, volume 2. Springer, 1999.

Gareth O Roberts and Jeffrey S Rosenthal. Optimal scaling of discrete approximations to langevin diffusions. Journal of the Royal Statistical Society: Series B (Statistical Methodology), 60(1):255–268, 1998.

Roman Schefzik, Thordis L Thorarinsdottir, and Tilmann Gneiting. Uncertainty quantification in complex simulation models using ensemble copula coupling. 2013.

Matthias W Seeger, Christopher KI Williams, and Neil D Lawrence. Fast forward selection to speed up sparse gaussian process regression. In International Workshop on Artificial Intelligence and Statistics, pp. 254–261. PMLR, 2003.

B. Shahbaba, S. Lan, W.O. Johnson, and R.M. Neal. Split Hamiltonian Monte Carlo. Statistics and Computing, 24(3):339–349, 2014.

Jiawei Su, Danilo Vasconcellos Vargas, and Kouichi Sakurai. One pixel attack for fooling deep neural networks. IEEE Transactions on Evolutionary Computation, 23(5):828–841, 2019.

Vivienne Sze, Yu-Hsin Chen, Tien-Ju Yang, and Joel S Emer. Efficient processing of deep neural networks: A tutorial and survey. Proceedings of the IEEE, 105(12):2295–2329, 2017.

Thordis L Thorarinsdottir and Tilmann Gneiting. Probabilistic forecasts of wind speed: Ensemble model output statistics by using heteroscedastic censored regression. Journal of the Royal Statistical Society Series A: Statistics in Society, 173(2):371–388, 2010.

M. Welling and Y.W. Teh. Bayesian learning via stochastic gradient langevin dynamics. In Proceedings of the 28th International Conference on Machine Learning (ICML), pp. 681–688, 2011a.

Max Welling and Yee W Teh. Bayesian learning via stochastic gradient langevin dynamics. In Proceedings of the 28th International Conference on Machine Learning (ICML-11), pp. 681–688, 2011b.

Scott Cheng-Hsin Yang, Wai Keen Vong, Ravi B Sojitra, Tomas Folke, and Patrick Shafto. Mitigating belief projection in explainable artificial intelligence via bayesian teaching. Scientific reports, 11(1):9863, 2021.

C. Zhang, B. Shahbaba, and H. Zhao. Hamiltonian Monte Carlo acceleration using surrogate functions with random bases. Statistics and Computing, 27(6), 2017a. ISSN 15731375. doi: 10.1007/s11222-016-9699-1.

C. Zhang, B. Shahbaba, and H. Zhao. Hamiltonian Monte Carlo acceleration using surrogate functions with random bases. Statistics and Computing, 27(6), 2017b. ISSN 15731375. doi: 10.1007/s11222-016-9699-1.

C. Zhang, B. Shahbaba, and H. Zhao. Variational hamiltonian Monte Carlo via score matching. Bayesian Analysis, 13(2), 2018. ISSN 19316690. doi: 10.1214/17-BA1060.

Cheng Zhang, Babak Shahbaba, and Hongkai Zhao. Hamiltonian monte carlo acceleration using surrogate functions with random bases. Statistics and computing, 27:1473–1490, 2017c.

