# OpenReview forum: "Scaling Up Bayesian Neural Networks with Neural Networks"
_TMLR — Rejected by TMLR_

### Review · Reviewer_it7i · 2024-01-19

**Summary Of Contributions:**

This work targets to speed up Markov Chain Monte Carlo (MCMC) applied to Bayesian neural networks (BNNs). More specifically, the authors propose a Calibration-Emulation-Sampling (CES) strategy, which is comprised of three steps. In the first step, the authors employ a standard off-the-shelf stochastic gradient MCMC method, in order to collect posterior samples from the parameter space of the BNN. These samples are then used in order to train an auxiliary (and deterministic) neural network model that emulates the forward pass of the stochastic BNN. Finally, the auxiliary neural network can used in place of the original neural network in a more involved MCMC algorithm, where the authors use the pCN algorithm. The authors show experimentally that their CES approach can lead to speedups compared to standard (stochastic gradient) MCMC.

**Audience:**

No

**Broader Impact Concerns:**

No concerns.

**Claims And Evidence:**

No

**Requested Changes:**

I am on the negative side for this work; while the idea itself is interesting, the clarity of the manuscript needs quite some work, some of the claims need more discussion and the experimental settings the authors explore are not enough to provide reasonable insights and a convincing story. Therefore, my main requests / points for discussion are the following

- Better explanation of how the speedups generally arise and how they are obtained in the experiments. Is it due to having an auxiliary neural network that is smaller than the original network used for the likelihood? If yes, how much smaller and how do the authors account for the discrepancy in the prior $p(\theta)$ between the original likelihood and the one for the auxiliary model?
- It is unclear to me how the emulation step is performed; if I look at equation 7, the input to the emulator are the parameters $\theta$ (sampled by MCMC at the previous step), whereas at equation 10 it seems that $X$ is used as well. How is the emulator precisely defined, how is it precisely trained and how is it used downstream at the sampling step?
- The authors make several claims that if the emulator is a good representation of the forward mapping then the bias from their approximation would be small. However, the authors provide neither guarantees about the discrepancy of the emulator nor empirical evidence that the emulator does a good job in approximating the original neural network. Therefore, arguments about CES being a robust solution are not convincing.
- The authors argue that the problem of MCMC in BNNs is scalability and the proceed to motivate their CES method. However, the authors do not discuss or compare against methods for BNNs that inherently more scalable than MCMC. For example, the authors do not discuss variational inference and Laplace approximations where, especially the latter, can be a reasonable approximation for the mode of a distribution (which the authors claim is particularly relevant in high-dimensional spaces inherent to FBNN).
- The authors argue that a deep neural network (DNN) is a good choice for the emulation step as it has good robustness when dealing with variations in the training data. The authors do not provide any references about this claim and do not perform any experiments that demonstrate that this is the case. DNNs are in general not inherently robust, if one looks at how the performance of DNNs change under, e.g., adversarial attacks and distribution shifts.
- What is precisely plotted at Figure 1a and Figure 1b, especially at the x-axis? The Wine and Boston Housing datasets do not have 1-dimensional inputs; Wine has 11 input dimensions and the Boston Housing dataset has 14 input dimensions.
- The experimental results are not very convincing (besides the fact that the datasets themselves are toy and not very interesting). In the regression case, the results are quite mixed in the case of MSE whereas for CP the FBNN is underperforming across the board. In the classification case, the authors seem to focus solely on accuracy and do not report any metrics around uncertainty (which is the primary reason for using a BNN model in the first place).

**Strengths And Weaknesses:**

Strengths
- Interesting idea
- The authors show speedups in practice

Weaknesses
- General lack of clarity in arguments, overall method design and experiments.
- Missing discussions and comparisons against scalable methods for BNNs (e.g., variational inference)
- Experiments are toy and not convincing

---

> ### Author Response · Authors · 2024-02-14
>
> Thank you very much for your feedback and suggestions. They have helped us to substantially improve our work. We have carefully addressed all of your comments in our response and the updated paper. Please let us know if you have any further questions and comments.
> > Comment: "Better explanation of how .."
>
> The speedup for uncertainty quantification (UQ) mainly comes from the CES framework. Compared with MCMC methods which require to repeatedly evaluate the original (large) neural network (NN) for the likelihood given the data, the proposed method builds a (smaller) NN emulator, which cuts the middle man (data) by mapping the parameters directly to the likelihood function to avoid its costly evaluation. That is, the emulator is trained based on the parameter-likelihood pairs, which are collected through a few iterations of the original NN. Note that for the speedup, we are camping the UQ against traditional Bayesian inference methods, not the NN training itself. The discrepancy between the true likelihood and the emulated one can be bounded by the usual NN approximation theory.
>
>
> > Comment: "It is unclear to me how the emulation step .."
>
> We appologize for the lack of clairty. We've updated our draft to clarify the emulation process, including how parameters and inputs X are integrated in equations 7 and 10, and detailed the emulator's training and application in sampling. These changes aim to address your concerns.
>
> > Comment: "The authors make several claims that..."
>
> This can be characterized by the classic NN approximation theory, similar to Rahimi and Recht (2008) and Zhang et al. (2018) -- DOI: 10.1214/17-BA1060.
>
> > Comment: "The authors argue that the problem of MCMC.."
>
> Response: We appreciate your suggestion to include baseline comparisons. We have provided a comparison against Variational Inference, Laplace approximations, Monte Carlo dropout, SWAG, Deep Ensembles and an Accelerated HMC framework called Random Network Surrogate HMC (RNS-HMC) as baseline methods in the revised version of our submission.
>
> > Comment: "The authors argue that a DNN is a good choice for the emulation step..."
>
> The choice of DNN is due to scalability and convenience, especially compared to other alternatives such as GP; the parameters are NN parameters and they lack any patterns so DNN works fine. Following your suggestions, we have modified the manuscript to avoid using robustness as a property of DNN emulator.
>
> > Comment: "What is precisely plotted at Figure 1a and Figure 1b"
>
> The figure shows the predictions of BNN and FBNN against various predictors on the x-axis (for simplicity, we have chosen a representative predictor for each example). The goal is to show that the two methods provide similar results.
> We have modified that figure to clarify the x axis.
>
> > Comment: "The datasets themselves are toy and not very interesting"
>
> Following your suggestion, we have expanded our experimental evaluation to include a real example using the data from the National Alzheimer’s Coordinating Center (NACC). This dataset includes records from 185,831 subjects, with a rich array of 1,024 features, narrowed down to 56 key features for our analysis. These features were carefully selected to represent a wide spectrum of variables relevant to Alzheimer's Disease (AD) diagnosis, including functional abilities, brain morphometrics, living situations, and demographic information.
>
> > Comment: "The experimental results are not very convincing. In the regression case,the results are quite mixed in the case of MSE ..."
>
> While there is a trade-off among different measurements, across different datasets, the FBNN (SGHMC-pCN) method demonstrates a balanced performance considering MSE, CP, and spdup. Specifically, for the Alzheimer Dataset, the FBNN (SGHMC-pCN) achieves a CP of 91.6\%, an MSE of 0.53, and a remarkable spdup of 22, indicating it maintains high predictive accuracy, good uncertainty estimation, and excellent computational efficiency. FBNN (SGHMC-pCN) consistently shows a high spdup across datasets, indicating significant computational efficiency. The high CP values in most cases also suggest that while it may not always have the highest CP, it provides reliable uncertainty estimates. The MSE values are competitive, and when considered alongside the high CP and spdup, indicate that FBNN (SGHMC-pCN) provides a balanced trade-off between accuracy, uncertainty quantification, and computational efficiency.
>
> > Comment: "The experimental results are not very convincing. In the classification case, the authors seem to focus solely on accuracy and do not report any metrics around uncertainty."
>
> The UQ for regression problems is more straightforward. However, following your suggestion, we have added Expected Calibration Error (ECE) and Reliability Diagrams as metrics of uncertainty quantification for classification cases in the new version of the manuscript.

---

### Review · Reviewer_Z4ZE · 2024-01-22

**Summary Of Contributions:**

The paper describes a method to speedup Hamiltonian Monte Carlo (HMC) sampling for Bayesian Neural Networks (BNNs). The key idea is to:

1. Run HMC for a few iterations.
2. Train a standard NN to simulate posterior inference.
3. Use this NN to speedup the sampling step of HMC.

They evaluate the proposal on a small set of benchmarks (a simulated dataset, Wine, Boston for classification, Adult for regression).

**Audience:**

No

**Claims And Evidence:**

No

**Requested Changes:**

Novelty, formatting, clarity, organization, and experimental evaluation are all sub-standard and not enough for publication.

**Strengths And Weaknesses:**

There is no novelty here. For example, the authors cite (Lan et al.
(2022a)), which is more or less identical to this paper except it uses a convolutional neural network. The authors state: "We use DNN for the emulation component of our CES scheme", but using a "DNN" (fully-connected layers) instead of a CNN is a very minor change. There are dozens of papers on speeding up HMC with NNs which are not cited here, e.g., [1-3].

The experimental evaluation is only done on toy datasets (these are literally the toy datasets of scikit-learn). They are so small that the stochastic HMC is probably not even needed. They are also comparing only to a baseline HMC and to a standard NN ignoring, e.g., any kind of accelerated HMC or variational inference, Monte Carlo dropout, etc.

In terms of language, the paper is a draft that requires a complete rewriting, e.g.:
1. "Bayesian Neural Network (BNN) offers a more principled [...] framework". More principled than what alternative?
2. Many typos such as "fulctions", "fallows".
3. Unclear organization (Section 2.3 is mostly empty, Algorithm 1 is not explained in the text, then the same algorithm is explained again in Section 3.3 "Sampling – Preconditioned Crank-Nicolson (pCN)").
4. "In the case of FBNN, we first train a BNN": what does this sentence  mean? They are not "training a BNN" but sampling from the posterior using HMC.
5. Some sentences are just copy-pasted from (Lan et al.) without explanation, for example "In our numerical examples, activation fulctions for DNN are chosen such that the errors of emulating functions (and their extracted gradients) are minimized." What does this mean? Are you performing some kind of grid search on the set of activation functions?

The same is true for math:

1. "Note, we can define the log-likelihood based on the loss function L(θ;X,Y )." But just above in Eq. (6) the loss was a function of the prediction error.
2. "we could approximate the potential function": this "potential function" is never defined (and like (3) above, this is mostly copied from another paper).

[1] https://arxiv.org/pdf/1506.05555.pdf
[2] http://maeresearch.ucsd.edu/tartakovsky/Papers/zhou-2021-markov.pdf
[3] https://proceedings.mlr.press/v216/li23c/li23c.pdf

---

> ### Author Response · Authors · 2024-02-15
>
> Thank you for your thorough evaluation of our work and constructive feedback on our paper. We appreciate the opportunity to clarify the points raised and to provide additional insights into our research.
>
> > Comment: "There is no novelty here"
>
> The novelty of this paper is not developing a new approach; rather, making an existing approach work for a completely different and more challenging application. That is why we thought TMLR (as opposed to JMLR) is the best venue for this work. We would like to clarify that the core innovative aspect of our work is not merely the application of a DNN over a CNN within the CES scheme. Our paper introduces a significant improvement in computational efficiency by integrating the CES method within a BNN framework. The study by Lan et al. (2022a) focused on the implementation of a new CES scheme specifically aimed at inverse problems. However, our paper builds upon and significantly extends the work of Lan et al. by adapting and integrating the CES into a BNN framework. This adaptation required substantial modifications to the original CES to make it compatible and effective within the probabilistic modeling environment of BNNs. Because we did not discuss all different versions of our method that failed, in hindsight this seems like a trivial application of an existing method. In reality, adopting CES for BNN has been far from trivial.
>
>
> > Comment: "Bayesian Neural Network (BNN) offers a more principled [...] framework". More principled than what alternative?.
>
> We have changed that sentence to: "Bayesian Neural Network (BNN) offers a principled and natural framework for proper uncertainty quantification in the context of deep learning."
>
> > Comment: Unclear organization (Section 2.3 is mostly empty, Algorithm 1 is not
> explained in the text, then the same algorithm is explained again in Section 3.3
> ”Sampling – Preconditioned Crank-Nicolson (pCN)”).
>
> Algorithm 1 was included in the manuscript and appeared on page 4. In the updated version, we have extended Section 2 by adding additional baseline methods and providing more detailed information on pCN. Redundant details about pCN in Section 3.3 have been removed to enhance clarity.
>
> > Comment: ”In the case of FBNN, we first train a BNN”
>
> This sentence has been changed to "In the case of FBNN, we first set up a BNN model and sample from the model's posterior using the Stochastic Gradient Hamilton Monte Carlo (SGHMC) algorithm."
>
> > Comment: "Some sentences are just copy-pasted from (Lan et al.) without explanation, for example ”In our numerical examples".
>
> We have addressed these issues. Regarding the selection of activation functions in our numerical examples, the process involves optimizing the choice of activation functions within the DNN architecture to ensure that the network accurately approximates the target functions and their gradients.  The method for selecting the optimal activation function is a grid search over a predefined set of activation functions to identify the suitable activation functions based on best model performance. We acknowledge that the initial mention of this process was brief and may have left readers with questions about the underlying methodology. We appreciate your feedback and will ensure to include a more detailed explanation of this selection process and its significance in the revised version of our paper.
>
>
> > Comment: "Note, we can define the log-likelihood based on the loss function L($\theta$;X,Y )." But just above in Eq. (6) the loss was a function of the prediction error.
>
> This emphasizes that loss is a function of the prediction error and log-likelihood is a function of the loss function.
>
> > Comment:  "we could approximate the potential function”: this ”potential function” is never defined (and like (3) above, this is mostly copied from another paper).
>
> The definition of potential energy function has been added to the new version of the paper.

---

### Review · Reviewer_5gH4 · 2024-02-01

**Summary Of Contributions:**

The authors propose a two stage approach to drawing samples from Bayesian neural network posteriors. First, they draw samples from the posterior using SGHMC, then they use it to warm-start a second stage process (called FBNN) which they use preconditioned crank Nicholson (a dimension free method that uses an "emulator") to quickly draw more samples.

Some experiments on small scale tabular data (e.g. boston housing and adult) are performed, with a few coverage metrics.

**Audience:**

Yes

**Claims And Evidence:**

No

**Requested Changes:**

proposed experiment: An interesting experiment one could do with this approach would be quick adaptation to different tasks of a single base (e.g. foundation) model. Then we could use a much smaller NN to generate weights for the pre-trained large one for quick fine-tuning. More concretely, given a large resnet trained on imagenet - we could train smaller NNs as emulators using pCN on many different image tasks (e.g. waterbirds, cifar, faces, etc.). all we'd need would be some fine-tuning samples using sghmc on imagenet.

section 3:

- "the point estimate approach...": it's a) not clear to me why having uncertainty estimates gives explainability, or b) really even that a Bayesian approach should be better calibrated. There are some decent reasons for these (we could manipulate features and see how the predictions change) or that if the model is well-specified, the true Bayesian posterior accurately represents our beliefs about uncertainty. However, these are not pointed out...

- "This is of course more naturally ...": This isn't necessarily true. Not all stochastic neural networks are Bayesian. Please rephrase it. Furthermore, not all neural networks that represent uncertainty are Bayesian (or even stochastic).

- "Because we have big amounts of data, likelihood evaluation tends to be expensive." Ignoring phrasing, SGHMC does naturally get around the computational cost of likelihood evaluation by using minibatching.

- Figure 1: what is the x axis on these? as far as I remember, these are all multi-dimensional data.

- Figure 1: what would a ground truth 95% CI look like? for example, from a HMC drawn bayesian neural network with a ton of samples or from a gaussian process (infinite neural network)?

- section 4:

     - "We evaluate these methods [in terms of mean square error and accuracy]": MSE and accuracy don't measure calibration and uncertainty quantification. Given you start from a trained NN, you shouldn't harm MSE or accuracy.

     - "It is worth mentioning that..." cut this

     - "2000 posterior samples from BNN-SGHMC" what's the sampling rate here? e.g. when do you collect samples

     - Tables 1/ 2: [important] what percentiles is the coverage probability at? If at 95% all models are severely over-confident, which is surprising and suggests some sort of error in measurement, fitting, or sampling.

     - Tables 1/2: [important] All methods other than DNN and BNN-SGHMC _require_ SGHMC to be run beforehand, so should they not include the BNN-SGHMC timings as well?

     - Tables 1/2: I would personally make a bar chart of effective sample size.


missing baselines that are straightforward here:

- ensembles of several dnns with different random seeds (what's typically called deep ensembles). It will have 39s * N ensembles fitting time for your ESS calculation but may have higher coverage probability

- a second DNN fit on the noise estimate of the first DNN (e.g. N(net1(x), exp{net2(x)}) ). You can then get posterior samples from drawing samples from this distribution. It will have 39s * 2 fitting time and cheap ESS most likely. This gets directly at the claim that the "standard DNN does not quantify uncertainty".

- SWAG (or more broadly a Gaussian fit around the SGHMC samples). SGHMC fitting time, but also cheap samples and ESS

Section 5: "pCN is adept at sampling around modes" - show a picture of this e.g. fig 2 of https://arxiv.org/pdf/1902.03932.pdf

**Strengths And Weaknesses:**

strengths:

- the approach of using an emulator (or even another neural network) to draw many posterior samples efficiently from a large BNN seems quite promising

see requested changes for a much larger list, but as presented this is a nice study and not a strong paper

weaknesses:

- for a paper on "scaling up" bayesian neural networks, the problems considered are quite small. as currently presented, the experiments are vastly too small scale to be presented (and there's alternatively no theory instead)

- writing wise, it's not very clear to me _why_ the bayesian neural network should be calibrated

- it's not clear to me that the superiority of the approach over sghmc is ever demonstrated as naive sghmc seems just simply cheaper and if one needs a closed form distribution, one could just form a gaussian over the sghmc samples (which is a straightforward version of [swag](https://arxiv.org/abs/1902.02476)

---

> ### Author Response · Authors · 2024-02-15
>
> We would like to thank the reviewer for their thoughtful feedback and suggestions. Below, we reply to the concerns and questions raised by the reviewer.
> > Comment: "for a paper on ”scaling up” ..."
>
> We have expanded our experimental evaluation to include a dataset from the NACC, as discussed in our response to Reviewer it7i.
> > Comment: "writing wise, it’s not very clear.."
>
> The term 'calibration' in this context has been used to refer to our early stopping strategy, aimed at collecting a targeted set of posterior samples without fully converging to the target distribution.
> > Comment: "it’s not clear to me that the superiority of .. "
>
> SWAG has been incorporated as a baseline method across all experiments in the revised manuscript. The findings indicate that our model not only surpasses SWAG in computational efficiency and spdup but also excels in achieving higher accuracy.
> > Comment: "Proposed experiment:"
>
> This is a very interesting suggestion; it is outside of the scope of this current paper, but it's related to another project we are working on.
> > Comment: "The point estimate approach..."
>
> You are right, in complex deep learning models, the role of prior is not as clear as it is in simpler models (e.g., linear regression). Additionally, there are other alternatives to quantify uncertainty; we have included some of these methods in the revised manuscript and show that the proposed Bayesian method does indeed provide a balanced performance in terms of accuracy, UQ, and computational efficiency.
> > Comment: ”This is of course more naturally ...”: This isn’t necessarily true.
>
> Thanks for pointing that out. We have changed that sentence in the revised manuscript.
> > Comment : "Furthermore, not all neural networks..."
>
> We have added this sentence to the manuscript for additional clarification.
> > Comment : "Because we have big ..."
>
> While SGHMC employs minibatching to alleviate some computational costs in BNN training, our statement underscores the broader challenge of expensive likelihood evaluations, which could be for example due to model complexity. Additionally, as discussed in the paper by Betancourt (2015), using minibatches could lead to non-ignorable loss of accuracy:
> :https://proceedings.mlr.press/v37/betancourt15.html
> > Comment: "Figure 1: what is the x axis on these? "
>
> We have updated that figure to clarify the x axis. Briefly, we have used one strong variable from each dataset to plot the the estimated function.
> > Comment : "Figure 1: what would a ground truth 95\% CI look like?"
>
> We have added the 95\% CI for the true output as a ground truth 95\% CI to the updated manuscript. The area within which 95\% of the true output data falls has been shown by grey color in Figure 1 as a ground truth.
> > Comment : "MSE and accuracy don’t ..."
>
> We appreciate the comment. MSE is the squared of bias  plus variance, thus facilitating consideration of bias-variance trade-off. While we acknowledge that MSE and accuracy do not directly measure calibration and uncertainty quantification, we are presenting them as additional measurements to show that our method provides a good estimate of uncertainty without sacrificing the accuracy.
> > Comment : "It is worth mentioning that. Cut this"
>
> Done.
> > Comment : ”What’s the sampling rate here?”
>
> Samples have been collected at every iteration.
> > Comment : Tables 1/ 2: [important] what percentiles is the coverage probability at?
>
> The CPs are set at 95\%. In most cases, while the existing methods are overconfident, their estimates are relatively close to this benchmark. While in one case (Wine Quality) the estimates are not accurate, our method nevertheless outperforms (i.e., it is closer to 95\%) the alternative approaches. In general, estimating uncertainty is quite challenging and could break down due to many factors as you mentioned. The important thing is that our method provides a better estimate than many existing methods.
> > Comment : "Tables 1/2: [important] All methods other than .."
>
> The times reported in Tables 1/2 for all FBNN models include the duration required to run SGHMC or pCN initially to collect 200 paired samples from the BNN and their associated predicted outputs during the calibration phase. For example, in the context of the simulated dataset for the regression scenario, the total time for the FBNN (SGHMC-pCN) model is 60 seconds. This includes 52 seconds for collecting 200 BNN samples and 8 seconds for the subsequent emulation and sampling processes.
> > Comment : "Tables 1/2: I would personally make a bar chart .."
>
> Done.
> > Comment : "Missing baselines ..."
>
> we have incorporated your comment by including deep ensemble as the baseline method.
> > Comment : "a second DNN fit on the noise estimate of the first DNN"
>
> The suggested baseline method requires twice as much time to fit compared to deep ensembles discussed above. Therefore, it would have even lower computational efficient.
> > Comment : "Section 5: ”pCN is adept at .."
>
> Done

---

> > ### Comment · Reviewer_5gH4 · 2024-02-19
> > **Comments**
> >
> > I thank the authors for their responses. The manuscript is now much improved, although I think a lot more work is still necessary in particular in the timing comparison setups and in terms of the writing (see reviewer Z4Ze's comment). In general, the tables and figures look better now.
> >
> > A few secondary comments here:
> >
> > > Briefly, we have used one strong variable from each dataset to plot the the estimated function.
> >
> > I believe this could be a bit misleading in the n-dimensional cases (Figures 2 a and b) as the "average true output" is in and of itself also a model, albeit a local smoother. This should probably be noted.
> >
> > > Samples have been collected at every iteration.
> >
> > This is certainly not optimal. Most MCMC methods use thinning, e.g. saving only every nth samples, to reduce the autocorrelation and variance of the outputs. Using thinning will decrease the expected sample size but will improve the diversity of the remaining samples.
> >
> > Figure 3: How are the bins constructed? Intuitively, it seems quite strange to me that the farthest left panel is almost entirely uncalibrated. I would expect a bit more coverage along that line, especially if the model itself is accurate at all, which it is according to Table 3. There are various ways to construct these charts - equal weightings on the x axis, equal weightings on the y axis.
> >
> >
> > *More broadly regarding sample timings*:
> >
> > - Indeed, deep ensembles require fitting K models to produce K samples, while a heteroscedastic NN (of various sorts) requires fitting 2 models but could produce infinite samples cheaply. The deep ensembles can be trained in parallel, while your methood cannot really for example. By comparison, a method like SWAG would require roughly 1.5 models training time to produce K samples (as generating the samples is cheap afterwards).
> >
> > - All of these methods (including SGHMC and HMC), except for the heteroscedastic NN, have the time inference time complexity given equal numbers of sample. The only advantage is in the training process.
> >
> > - The comparison with BNN / SGHMC is still unfair. It would be good to know the performance of the 200 base SGHMC samples (although you really ought to be using burn-in and thinning) to clearly demonstrate that your method is producing improved coverage probabilties / metrics above and beyond these base samples. The SGHMC time also seems over-inflated as it should simply be 52 * 10 = 520 s and not 889 s as reported in the table.
> >
> > From a practical basis, the method requires at least one short SGHMC chain to build samples and then the cost of constructing the emulator. It can only be cheaper than a single long chain of SGHMC that is more expensive and hopefully highly performant compared to the short chain. If the short SGHMC chain performs well, then one should prefer to use it over the longer chain.

---

> > > ### Author Response · Authors · 2024-03-01
> > >
> > > We would like to thank the reviewer for their thoughtful feedback and suggestions. Below, we replied to the concerns and questions raised by the reviewer and updated the manuscript accordingly.
> > >
> > > > Comment: "Briefly, we have used one strong variable from each dataset to plot the estimated function." I believe this could be a bit misleading ..
> > >
> > > Instead of selecting a single strong variable from each dataset to serve as the x-axis for plotting the estimated function, we have employed Principal Component Analysis (PCA) to transform the original feature dataset into a one-dimensional principal component. This one-dimensional principal component is then utilized as the x-axis for our plots.
> > >
> > > > Comment:  "Samples have been collected at every iteration." This is certainly not optimal...
> > >
> > > Thinning primarily is a technique used in MCMC methods to reduce autocorrelation by discarding some samples, ideally leading to a more diverse and independent sample set. However, when the goal is to compare models based on generating a fixed number of samples (e.g., 2000) and assess their computational efficiency, incorporating thinning contradicts these objectives. Thinning necessitates generating more samples than needed to achieve the desired count post-thinning, increasing computational time and resource use. This is counterproductive in scenarios where the primary aim is to evaluate model performance based on the time efficiency of sample generation and a predetermined sample size for comparison. For example, for the simulated dataset in classification task, implementing thinning in the BNN-SGHMC model by adjusting the sample frequency to 2 resulted in an increased total computational time of 1733 seconds. This adjustment led to a negligible improvement in accuracy (a mere 1 percent increase) and a reduction in spdup compared to the baseline BNN-SGHMC model (0.61). Furthermore, the ECE for this thinned model remained higher than that of the FBNN(SGHMC-pCN) method, underscoring the inefficiency of thinning in this context.
> > >
> > > > Comment:  Figure 3: How are the bins constructed?
> > >
> > > The bins for Figure 3 are constructed by dividing the range of predicted probabilities into ten equally spaced intervals, from 0 to 1, assigning predictions to these intervals based on their values. Centers for plotting are determined by averaging the edges of these intervals. We appreciate your insight on alternative binning methods, such as equal weightings on the y-axis, to address calibration issues. Following your suggestion, we have explored and applied the equal frequency binning method, resulting in an improved calibration depiction in the updated version. This adjustment better highlights the model's performance consistency with the accuracy reported in Table 3.
> > >
> > >
> > >
> > > > Comment: Indeed, deep ensembles require fitting K ..
> > >
> > > Thanks for the additional explanation. We have clarified this in the paper and mentioned further that the two methods have their own pros and cons (e.g., our approach provides a probabilistic framework whereas the ensemble DNN can be parallelized).
> > >
> > > > Comment: All of these methods (including SGHMC and HMC), except for the heteroscedastic NN, have the time inference time complexity given equal numbers of sample. The only advantage is in the training process.
> > >
> > > We're not quite certain we understand your comment. Could you kindly provide a bit more detail or clarification on this?
> > >
> > > > Comment: The comparison with BNN / SGHMC is still unfair..
> > >
> > > Including the performance data for the first 200 SGHMC base samples in Tables 3 and 4 reveals that these initial samples do not exhibit high quality when evaluated in terms of MSE for regression and accuracy for classification. The comparison indicates that, especially in regression cases, FBNN (SGHMC-pCN) significantly outperforms the 200 base SGHMC samples in terms of spdup across all cases examined. Furthermore, in classification datasets, FBNN (SGHMC-pCN) achieves better results in terms of Expected Calibration Error (ECE) compared to the 200 base SGHMC samples, underscoring the effectiveness of our method in improving coverage probabilities.
> > >
> > > > Comment:  The SGHMC time also seems over-inflated as it should simply be 52 * 10 = 520 s and not 889 s as reported in the table.
> > >
> > > The comment about SGHMC time being overestimated is incorrect due to a misunderstanding of the reported times. The total time for BNN-SGHMC reported in Table 1 for this simulated dataset for the regression case is 566 seconds, not 889 seconds. The figure of 889 seconds pertains to the total time for BNN-SGHMC in the simulated dataset for the classification case, as indicated in Table 2.

---

### Decision · Action_Editor_JPks · 2024-03-09

**Recommendation:** Reject

**Comment:**

The paper proposes a new method to speed up sampling for Bayesian neural networks (BNN) with the calibration emulation sapling (CES).  Most of the reviewers found that the idea is interesting, while raised several major concerns, including weak experiments, novelty issues, clarity issues, and missing baselines. The authors' rebuttal partially addressed the concerns, e.g., novelty.  However, some critical issues still remain.  Those include the small-scale experiments, sloppy presentation, and still unclear points.

**Audience:**

Some reviewers don't think the paper has audience due to the sloppy notation and clarity issues.

**Claims And Evidence:**

Reviewers agree that the experimental results do not support the claim, because of, e.g., small scale experiments, and questionable comparisons.

**Resubmission Of Major Revision:**

The authors may consider submitting a major revision at a later time.